# Photosymbiosis shaped animal genome architecture and gene evolution as revealed in giant clams
Ruiqi Li [1,2,6] ✉, Carlos Leiva [3,6], Sarah Lemer[3,4], Lisa Kirkendale[5] & Jingchun Li[1,2]

Symbioses are major drivers of organismal diversification and phenotypic innovation. However, how long-term symbioses shape whole genome evolution in metazoans is still underexplored. Here, we use a giant clam (*Tridacna maxima*) genome to demonstrate how symbiosis has left complex signatures in an animal's genome. Giant clams thrive in oligotrophic waters by forming a remarkable association with photosymbiotic dinoflagellate algae. Genome-based demographic inferences uncover a tight correlation between *T. maxima* global population change and major paleoclimate and habitat shifts, revealing how abiotic and biotic factors may dictate *T. maxima* microevolution. Comparative analyses reveal genomic features that may be symbiosis-driven, including expansion and contraction of immunity-related gene families and a large proportion of lineage-specific genes. Strikingly, about 70% of the genome is composed of repetitive elements, especially transposable elements, most likely resulting from a symbiosis-adapted immune system. This work greatly enhances our understanding of genomic drivers of symbiosis that underlie metazoan evolution and diversification.

Photosymbiosis, wherein heterotrophic hosts establish relationships with photoautotrophic symbionts, has played an essential role in the evolution of life, shaping the ancient origins of the eukaryotic cell and organelles. It is also the foundation of coral reefs, one of the most biodiverse and productive ecosystems[1]. It involves efficient photosynthetic energy exchanges between autotrophs and heterotrophs, and can be found in a variety of animal hosts, ranging from sponges, cnidarians, to mollusks and amphibians[2]. Symbiosis requires complex crosstalk among the partners at multiple levels, such as chemical signaling, immune recognition, metabolic exchange, and host-symbiont population dynamics[3]. These interactions can trigger co-evolutionary dynamics among distantly related lineages[4], generate strong selection pressures on certain genomic regions[5], or relax pressure on other parts of the genome[6]. Growing work has started to uncover molecular mechanisms behind symbiosis through the lens of gene evolution. Such genes include putative pattern-recognition receptors involved in symbiont recognition, as well as transporters for carbon, nitrogen, phosphorus, and trace metals that facilitate host-symbiont metabolic exchange[7–9].

However, how long-term symbioses shape whole genome evolution in metazoans is still underexplored[8,10,11]. Theoretically, symbiosis can alter genomes at multiple levels, including nucleotide substitutions, regulatory adaptations, transposon activity shifts, structure variations, and horizontal gene transfers[11–13]. Intriguingly, emerging evidence shows that photosymbiotic animal genomes appear to possess higher proportions of transposable elements compared to non-symbiotic relatives[8,11], suggesting that this underexplored aspect of genome architecture may play a significant role in maintaining symbiotic relationships or be influenced by symbiosis.

By forming a remarkable association with photosymbiotic dinoflagellate algae (Symbiodiniaceae), giant clams (subfamily Tridacninae) have secured their position as the largest bivalves on our planet. They possess the heaviest shells among all extant bivalves, with a recorded weight reaching up to 700 pounds. Their mantles display a captivating array of colorful patterns, establishing them as iconic inhabitants of coral reefs. Giant clams originated and diversified in the Indo-west Pacific in warm shallow tropical seas and have always been restricted to these environments[14,15]. Genetic evidence suggests that all giant clams are photosymbiotic, and this relationship is thought to have evolved once in their common ancestor at ~27 mya (SD = 4.4)[16], coinciding with the global expansion of modern coral reefs[17]. During this period, the emergence of shallow marine habitats dominated by other photosymbiotic organisms likely facilitated the evolution of photosymbiotic traits in giant clams by providing suitable environment and symbiont reservoirs. The genomic adaptations enabling Tridacninae to host photosymbionts likely lead to this subfamily's radiation

[1]Department of Ecology and Evolutionary Biology, University of Colorado Boulder, Boulder, CO, USA. [2]Museum of Natural History, University of Colorado Boulder, Boulder, CO, USA. [3]University of Guam Marine Laboratory, Guam, USA. [4]Centre for Molecular Biodiversity Research, Leibniz Institute for the Analysis of Biodiversity Change, Museum of Nature, Hamburg, Germany. [5]Collections and Research, Western Australian Museum, Perth, WA, Australia. [6]These authors contributed equally: Ruiqi Li, Carlos Leiva. ✉e-mail: Ruiqi.Li@colorado.edu

(6 mya[16]), allowing species and their populations to expand throughout the Indo-Pacific. These same innovations have also closely tied their demographic history and geographic distribution to the fate of coral reef ecosystems. Regrettably, like numerous other remarkable species, wild populations of giant clams face significant threats arising from overfishing and the adverse effects of climate change, which causes them to bleach[18]. In 1985, giant clam species were listed on Appendix II of the Convention on International Trade in Endangered Species of Wild Fauna and Flora (CITES).

In contrast to cnidarians that host photosymbionts intracellularly, giant clams harbor symbionts extracellularly, within an elaborate network of tubules derived from the digestive system[19]. The tubular system actively engages in transporting immobile symbiont cells across various organs, including the mantle and stomach, which is believed to maximize symbiont photosynthetic efficiency and regulate symbiont populations[20]. Several molecular mechanisms related to giant clam photosymbiosis have been proposed. For instance, in the tubular system, giant clams can promote symbiont photosynthesis by increasing supply of inorganic carbon through the V-type $H^+$-ATPase-dependent carbon-concentrating mechanism (CCM)[21]. Various transporters have also been identified that play a role in the nutrient exchange between the host and the symbionts, such as the sodium-dependent glucose transporter (SGLT1)[22] and the taurine transporter (SLC6A6), which transport exogenous taurine to stimulate photosynthate release[23]. In addition, light-induced production of signaling molecules in the symbionts may lead to increased enzymatic activity, which is essential for the host's calcification[24]. While substantial progress has been made regarding characterizing individual molecular pathways in giant clam photosymbiosis, a genomic-level comparative framework provides a comprehensive picture of the complexity of genomic signatures of photosymbiosis.

Here, we sequenced a high-quality reference genome for *Tridacna maxima*, a relatively small but overharvested giant clam species facing conservation challenges. Through comparative genomic analyses with congeners and other non-photosymbiotic mollusks, we demonstrate that its photosymbiotic ecology left unique genomic signatures and resulted in a genome that is distinct from non-symbiotic relatives in many aspects. These include a demographic history mirroring paleoclimate shifts and closely linked to that of coral reefs, a novel repertoire of genes related to immune functions, receptors and metabolism, and an extremely high proportion of transposable elements (Fig. 1).

## Results

### The chromosomal-level genome of *Tridacna maxima*

The 1.32 Gbp high-quality *Tridacna maxima* genome was produced using a combination of PacBio (~60X) and Dovetail Hi-C Omni-C libraries (~45X). The assembly contained 13,908 scaffolds with a N50 of 64 Mbp. 86% of the genome (1.14 Gbp) was encompassed in 18 putative chromosomes (Fig. 2A, Supplementary Fig. 1). Synteny plots show that the chromosome structure of *T. maxima* is consistent with its closely related species *T. crocea*. However, *T. gigas*, a more distantly related congener, has one fewer chromosome, suggesting a possible chromosomal fusion or splitting event within the genus (Fig. 3).

We combined ab initio and transcriptome-based methods to predict genes in the assembly, obtaining a total of 46,469 gene models. Initial BUSCO assessment using the ab initio annotations appeared to be relatively low with 16.0% of missing BUSCOs (Supplementary Data 2). The final annotation adding the transcriptome-based gene models represented an improvement in the BUSCO assessment, increasing the completed and remarkably lowering the missing BUSCOs to 9.4%. Manual blastp of the missing BUSCOs decreased this value from 9.4% to 4.7%, as 12 out of the 24 missing BUSCOs could be manually found (Supplementary Data 2).

### Demographic history

Photosymbiosis traits in animals can influence their ability to survive or succumb to environmental changes including major geoclimatic events. We used the Pairwise Sequential Markovian Coalescent (PSMC) method[25] to estimate the effective population size ($N_e$) of *T. maxima* backwards in time, from 3 mya to 10 thousand years ago (kya) (Fig. 2B). The PSMC results showed a population expansion from ~3 mya, followed by a decline after ~1.3 mya at the onset of the Mid-Pleistocene Transition (MPT). A second steep decline coincides with the Last Glacial Period (LGP), from ~125 kya to ~10 kya.

### Gene characteristics

Molecular adaptation can be driven by variations in gene family sizes[26]. Thus, we conducted a comparative analysis of gene family evolution between *T. maxima* and 13 other non-*Tridacna* molluscan species. Our analysis revealed a total of 122 expanded gene families and 85 contracted gene families in the *T. maxima* genome (Supplementary Data 3, 4). Among these, only 38% of the expanded gene families can be confidently annotated with biological functions. Many of the expanded gene families are related to transposable elements (PiggyBac transposable element-derived protein, Reverse transcriptase domain-containing protein, SCAN domain-containing protein, etc.). These likely resulted from incomplete repetitive elements masking prior to gene model prediction, either because the sequences are too divergent from references or low copy numbers prevented their recognition as repetitive elements. Transposable elements aside, many of the contracted and expanded gene families had functions related to the immune system, and were pattern recognition receptors such as toll-like receptors (TLRs), thrombospondins, and P2X purinoceptors (Fig. 4A). To further validate that these expanded and contracted functional groups were not confined to only *T. maxima*, similar analyses were conducted using an additional two *Tridacna* species: *T. crocea*[27], which is closely related to *T. maxima*, and a more distantly related *T. gigas*[28]. The results revealed that similar functional groups, such as those related to immunity and metabolism, were expanded/contracted in all three *Tridacna* species (Fig. 4B, Supplementary Data 5, 6). In particular, many gene families related to the C1q/Tumor Necrosis Factor-Related Proteins (CTRP) are contracted (Fig. 4B).

We also examined genes and gene families unique to *T. maxima* (i.e., not found in other mollusk genomes), which revealed 1756 unique gene families and 5742 unique genes (Supplementary Data 7, 8). Excluding those that could not be annotated (54.3%), and those encoding proteins associated with transposons, major species-specific genes/gene families were involved in cell surface pattern recognition receptor signaling pathway, TOR signaling, and beta-tubulin binding. GO enrichment analysis highlighted several critical functions (Table 1, Supplementary Data 9). These included multiple transporters (e.g., L-proline transmembrane transporter, glycine transmembrane transporter, lipid transporter), shell formation related process and enzymes (e.g., calcium ion binding, chitin synthase), cytoskeletal motor activity, cilium-dependent cell motility, immunity, and apoptosis. Similarly enriched processes related to transportation, immunity, and metabolism are found in the genes and gene families uniquely shared by *T. maxima*, *T. crocea*, and *T. gigas* (Supplementary Data 1, Supplementary Data 10, 11). It is worth noting that these represented an underestimation of unique genes at the *Tridacna* genus level due to less extensive genome annotation of *T. crocea* and *T. gigas* – many genes might not be identified from these two genomes because they were annotated by limited RNA libraries[27,28].

Gene duplication analyses were conducted to characterize the nature of duplication for genes found in the expanded families. For *T. maxima*, most genes were dispersedly duplicated (3371 genes, 88.3%), followed by 248 (6.5%) genes in proximal duplications, 137 (3.6%) in tandem duplications, and 61 (1.6%) in segmental duplications. Similar patterns were found in expanded gene families shared by all three *Tridacna* species: 1032 genes (91.1%) were dispersedly duplicated, 38 genes (3.4%) in proximal duplications, 28 (2.5%) in segmental duplications, and 25 (2.2%) tandemly duplicated.

### Characterization and evolution of transposable elements (TE)

Genomic structure analysis revealed that the *T. maxima* genome has a notably higher proportion of repetitive elements (68.07% of its 1.32-Gb

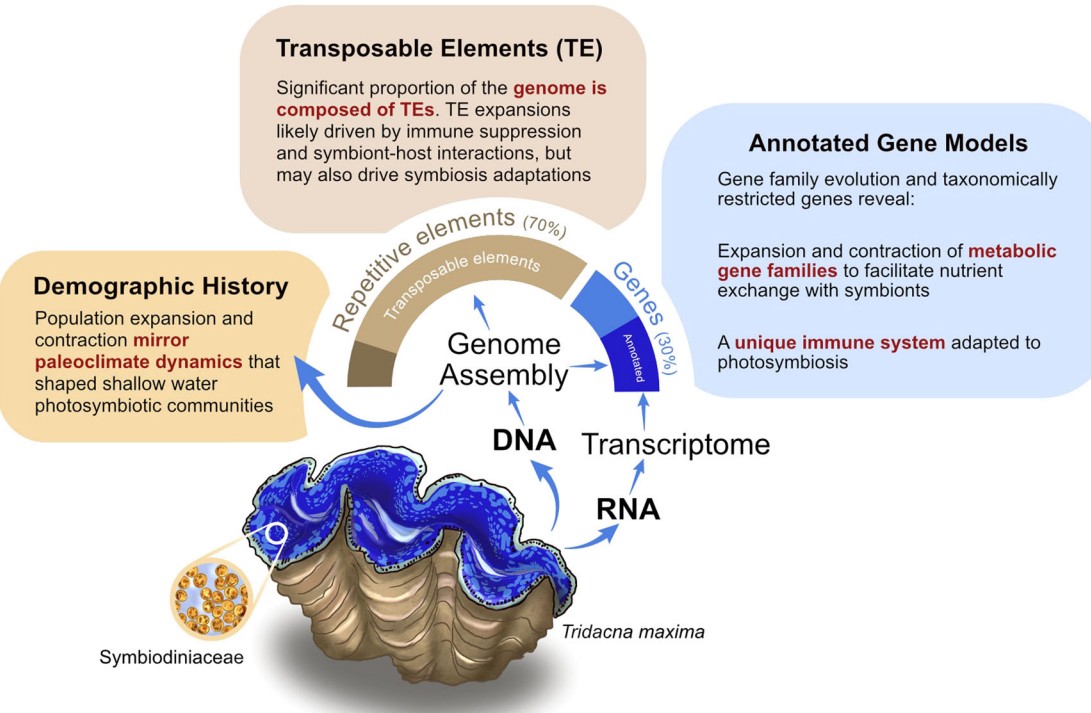

**Fig. 1 | Summary of major findings in this study.** Photosymbiosis ecology greatly impacted *Tridacna maxima* demographic dynamics, genome composition, and gene evolution.

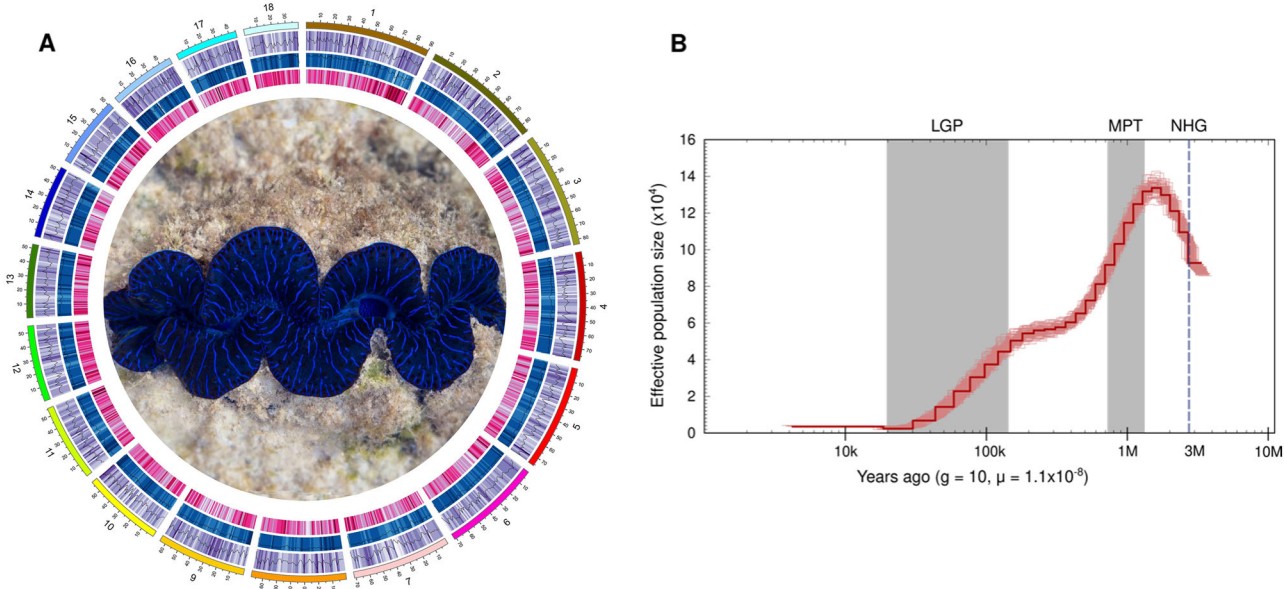

**Fig. 2 | Genomic characterization and demographic history of *Tridacna maxima*. A** Circos plot illustrating the characterization of the *Tridacna maxima* genome. From the outermost to the innermost circles: lengths of pseudo-chromosomes are represented in Mb; gene density is indicated in purple; transposable element (TE) density is shown in blue; and GC content is depicted in pink, all calculated in 1 Mb windows. Photo credit to Yu Kai Tan. **B** Demographic history inference of *T.* *maxima*. Effective population size ($N_e$) over time was estimated using the Pairwise Sequentially Markovian Coalescent (PSMC) model from the genomic pattern of heterozygosity. The dashed line indicates the northern hemisphere glaciation (NHG), while shaded areas represent the Mid-Pleistocene Transition (MPT) and the Last Glacial Period (LGP). g: Generation time (year). μ: mutation rate (per site per generation).

genome) compared to other bivalve species. For example, the 755.5-Mb assembled genome of a closely related cardiid *Cerastoderma edule* contains 37.81% repetitive elements[29] (Table 2). Given that the non-repetitive genomic regions in the two species are relatively similar in size (421 and 469 Mb respectively), it clearly points to repetitive elements as the main driver of larger genome size in *T. maxima*. The most abundant TE types are rolling-circle (RC) transposons (15.91%) and DNA transposons (23.26%), followed by SINE (Short Interspersed Nuclear Elements) (15.91%), LTR

**Fig. 3 | Synteny plots of three *Tridacna* species.**
Synteny plots of three *Tridacna* species: *Tridacna maxima*, *Tridacna crocea*, and *Tridacna gigas*. The synteny comparison reveals a possible chromosomal fusion or splitting event within the genus.

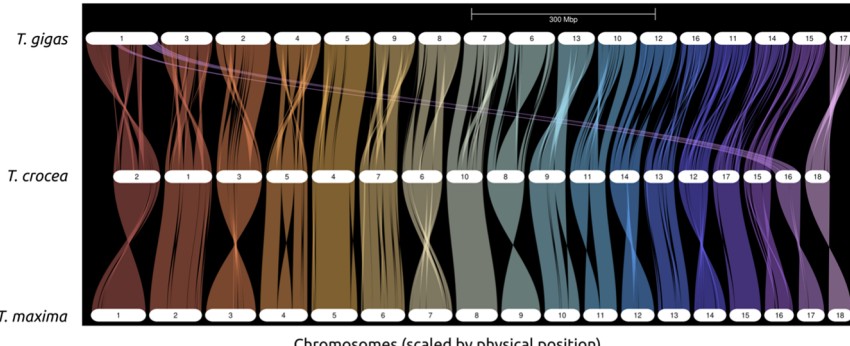

(Long Terminal Repeats) (2.4%), and LINE (Long Interspersed Nuclear Elements) (2.61%). Similar proportions of repeat elements were found in *T. crocea* and *T. gigas* as well (Table 2).

To explore the temporal dynamics of TE activity, insertion times of TEs were estimated by calibrating divergence rates from consensus TE sequences with calculated bivalve substitution rates[23]. Overall, TE insertion activities are very similar between *T. maxima* and *T. crocea*. In *T. gigas*, the overall trend is consistent for some TE types (RC, LINE, LTR) but deviates in DNA transposons and SINE (Fig. 5). LINEs, LTRs, and DNA transposons underwent rapid expansion during or shortly after the origin of photo-symbiosis in all three *Tridacna* species, but not in *C. edule*. SINEs and RC elements initially expanded rapidly in *Tridacna* at ~60 mya but gradually decreased. The same trends were not observed in *C. edule*.

## Discussion

*Tridacna maxima* is the widest ranging and most abundant member of the subfamily Tridacninae[30]. As a result, it could be considered the most successful of its kind. Our genomic study provides an assessment of how its photosymbiosis lifestyle influenced its evolution at multiple levels, in comparison to its congeners and other mollusks.

Genomes keep records of evolutionary and ecological forces that shaped populations and can be used to reconstruct the demographic history of a species[31]. This work provides the first genome-wide demographic study of a giant clam using PSMC, a coalescent-based approach. We showed that the photosymbiotic traits in *T. maxima* resulted in a demographic history closely linked to major geoclimatic events that also affected the growth and decline of shallow marine reef habitats. The Pliocene extinction of marine and terrestrial megafauna, triggered by Northern Hemisphere Glaciation (NHG)[32,33], likely led to a population expansion in *T. maxima* (effective population size ($N_e$) peak at 130k individuals) and other shallow water species like corals[34,35] and oysters[36] by freeing habitable niches. From 1.5 to 0.4 mya, both *T. maxima* and coral populations[37] was estimated to experienced a steep decline, potentially due to the Mid-Pleistocene Transition (MPT) and increased climate variability[34]. After stabilizing during the steady climate of the Mid-Brunhes Events (~0.4 Mya)[38], *T. maxima* populations was estimated to decline at the onset of the Last Glacial Period. This population pattern is once again mirrored in other shallow marine species, including photosymbiotic corals[34,35] that serve as habitats for giant clams.

The close relationship between the demographic histories of these two taxa (scleractinian corals and giant clams) and contemporaneous climatic events raises two observations: (1) The expansion of coral reefs around the Plio-Pleistocene boundary led to a major increase of shallow marine habitats that likely facilitated the expansion of *T. maxima*, as this provided suitable habitats for sustainable photosymbiosis. Intriguingly, the emergence of the currently most widespread giant clam-algal photosymbiont genera [*Cladocopium* (5 mya), *Durisdunium* (1.5 mya) and *Brevolium* (3 mya)][39] is estimated to have occurred around the same time, likely also contributing to the steep expansion of *T. maxima*. (2) The genomic acquisitions that lead to photosymbiotic adaptations in giant clams likely also made their survival tightly intertwined with that of coral reefs. As coral reefs are currently under

massive global and local anthropogenic pressure leading to mass mortality, the future of giant clams is uncertain[40–42]. The estimated declining trajectory of $N_e$, low genetic diversity and limited connectivity of contemporary *T. maxima* populations, compared to other bivalves[40–42], can drastically reduce their ability to adapt to rapid environmental changes, putting this well adapted species at high risk of extinction.

Our analyses revealed potential genomic basis related to giant clams' large and heavily calcified shells[43,44]. We found significant expansions in gene families that regulate calcification and growth in bivalves, such as the Calmodulin-A-like and the EGF-like domain-containing gene families[45,46]. Expansions of these biomineralization gene families might have driven diversification of gene functions and novelties responsible for giant clams' extraordinary growth rates and shell sizes[47].

The striking contraction of the C1q/Tumor Necrosis Factor-Related Protein (CTRP) gene family in *T. maxima* (and in the other two *Tridacna* species) may also be linked to its symbiont-fueled growth and ultimately large size. CTRP has been well studied in mammalian models because it plays a significant role in body weight control through reducing blood glucose and insulin levels. For example, CTRP9 knockout mice models exhibit increased body weight and impaired insulin resistance[48]. Giant clams derive substantial energy from their symbionts in the form of glucose[49]. Given that *Tridacna* receives constant glucose supply from algal photosynthesis, it is possible that the glucose-sensitive CTRP gene family is no longer beneficial and has been selected against to avoid interference with symbiont nutrient transfer. The loss of CTRPs may in turn lead to less prohibited weight gain and growth.

Our results also indicate that *T. maxima* genomic evolution has been greatly influenced by the need to balance immune responses, ensuring both the rejection of harmful pathogens and the maintenance of beneficial symbiotic partnerships. In the *T. maxima* genome, we observed significant expansions in many gene families, including several related to toll-like receptors (TLRs). TLRs are a type of transmembrane receptor known for their key role in innate immunity. They are crucial for identifying and defending against various microbial pathogens[50]. They are also known to support the recognition and maintenance of symbiotic relationships in other animal lineages[51], such as cnidarians[52] and tubeworms[53]. The consistent expansion of TLR in diverse symbiotic animals suggests that it has facilitated host-symbiont interactions. In addition to TLR, the expansion of other immune-related genes may reflect the crucial roles the immune system plays in the evolution of photosymbiosis. We observed expansions of the gene families of Inhibitor of Apoptosis (IAP) repeat-containing protein, which regulates apoptosis; techylectin, a key pattern recognition molecule in the innate immune response system; P2X purinoceptor, involved in innate immunity through the secretion of pro-inflammatory cytokines and induction of pyroptosis, and Thrombospondin type-1 repeat (TSR), promoting promotes colonization of the host[54–57]. Interestingly, certain immune system gene families, like some TLRs and Leucine-rich repeats, have undergone contractions. The complex evolution of immune system gene families in *T. maxima* hints at a highly sophisticated adaptation strategy, where the immune system is in part specialized in recognizing and

**Fig. 4 | Overview of expanded and contracted gene families in Tridacna species.** Overview of selected expanded and contracted gene families in **A** *Tridacna maxima* and **B** Three *Tridacna* species compared to thirteen other molluscan species, highlighting potential associations with symbiotic lifestyle in giant clams. OrthoDB gene family function and ID were listed for each gene family. For expanded gene families across the three *Tridacna* species, customed annotation and corresponding orthogroup number were listed. Images obtained from Adobe Stock with a standard license or from the National Museum of Natural History collections with permission.

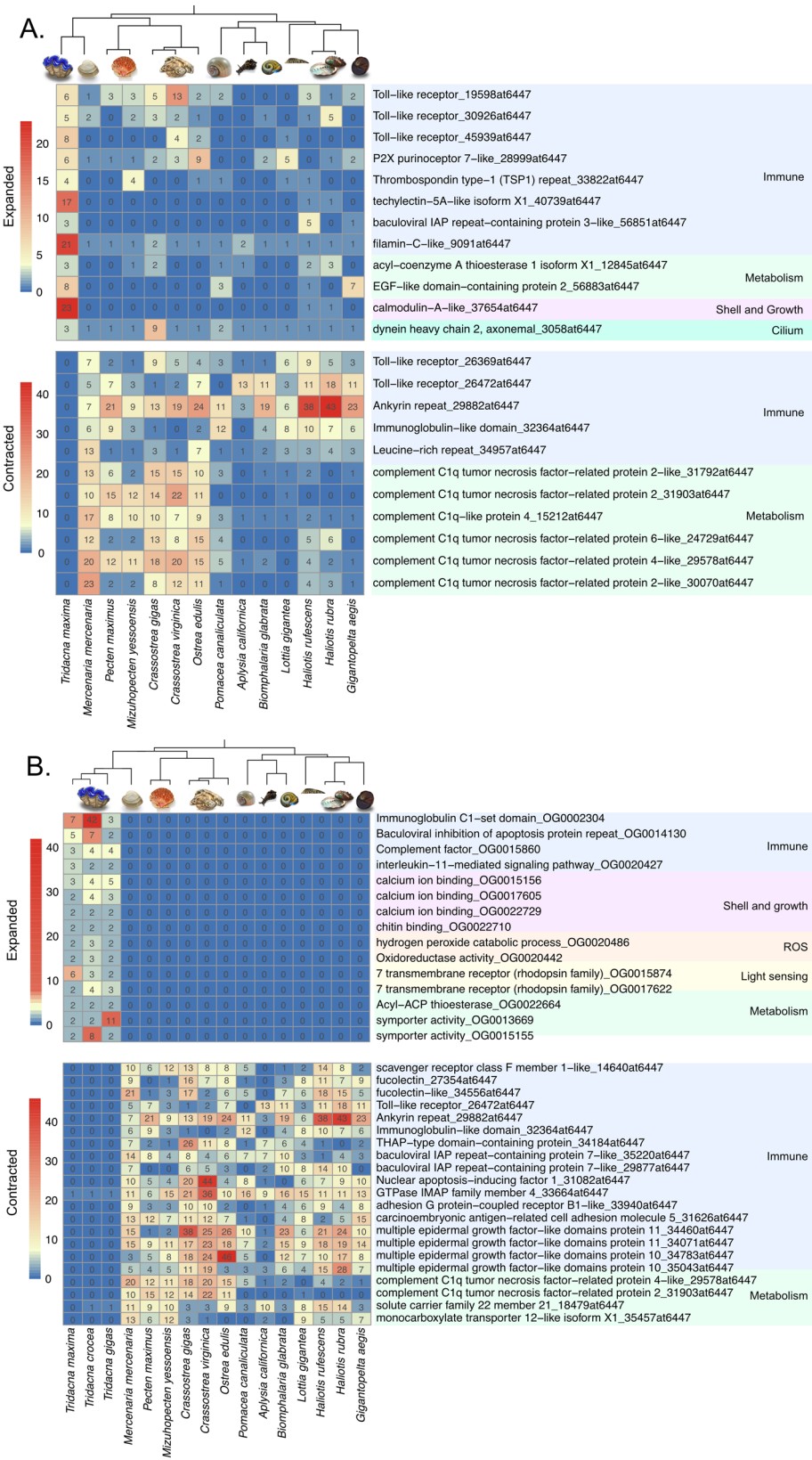

managing specific symbiotic partners (gene family expansions) while other aspects are being selectively suppressed (gene family contractions),likely facilitates the tolerance and long-term maintenance of symbionts by minimizing the host's immune reactions against them, and the selective downregulation of certain immune pathways could result in a compromised

ability to recognize and respond to pathogens. Although our inferences were mostly based on results from *T. maxima*, it is worth mentioning that similar patterns were found at the *Tridacna* genus level (when including *T. crocea* and *T. gigas*). Many immune gene families prevalent in other mollusks were completely missing in all three *Tridacna* species, and some immune-related

**Table 1 | Enriched gene ontology terms from taxon-specific genes and gene families in *Tridacna maxima***

| Function | Term | Name |
|---|---|---|
| Transporters | GO:0006811;GO:0034220;GO:0055085 | Monoatomic ion transport |
| | GO:0006820;GO:0015711 | Monoatomic anion transport |
| | GO:0015748;GO:0015914 | Organophosphate ester transport |
| | GO:0015804 | Neutral amino acid transport |
| | GO:0015816;GO:1903344;GO:1903346;GO:1903804 | Glycine transport |
| | GO:0015824;GO:0035524 | Proline transport |
| | GO:0005319 | Lipid transporter activity |
| | GO:0005548 | Phospholipid transporter activity |
| | GO:0015175 | Neutral L-amino acid transmembrane transporter activity |
| | GO:0015187 | Glycine transmembrane transporter activity |
| | GO:0015193 | L-proline transmembrane transporter activity |
| | GO:0140303 | Intramembrane lipid transporter activity |
| Shell formation | GO:0005509 | Calcium ion binding |
| | GO:0007591 | Molting cycle, chitin-based cuticle |
| | GO:0008363;GO:0042337;GO:0035293 | Cuticle development involved in chitin-based cuticle molting cycle |
| Cilium | GO:0003341 | Cilium movement |
| | GO:0007017 | Microtubule-based process |
| | GO:0007018 | Microtubule-based movement |
| | GO:0007021;GO:0007023 | Tubulin complex assembly |
| | GO:0051495;GO:1902905 | Positive regulation of cytoskeleton organization |
| | GO:0003774;GO:0003777 | Cytoskeletal motor activity |
| Immunity | GO:0006909 | Phagocytosis |
| | GO:0097190 | Apoptotic signaling pathway |
| | GO:0097193 | Intrinsic apoptotic signaling pathway |
| | GO:0097194 | Execution phase of apoptosis |
| | GO:2001233 | Regulation of apoptotic signaling pathway |
| | GO:2001235 | Positive regulation of apoptotic signaling pathway |
| | GO:2001242 | Regulation of intrinsic apoptotic signaling pathway |
| | GO:2001244 | Positive regulation of intrinsic apoptotic signaling pathway |
| | GO:1902229 | Regulation of intrinsic apoptotic signaling pathway in response to DNA damage |
| | GO:0043903 | Regulation of biological process involved in symbiotic interaction |
| | GO:0050776 | Regulation of immune response |
| | GO:0050778 | Positive regulation of immune response |
| | GO:0002684 | Positive regulation of immune system process |
| | GO:0006955 | Immune response |

Selected enriched GO terms from taxon-specific genes and gene families that are unique to *Tridacna maxima*, indicating potential functions related to the symbiotic lifestyle and large size of giant clams.

genes share within *Tridacna* were not found in any other mollusk genomes included in this study.

Taxonomically restricted genes (TRGs) are genes unique to certain groups of organisms[58]. Recent genomic research spanning the entire tree of life has increasingly demonstrated that TRGs often play a pivotal role in the development of unique phenotypes for example in *Hydra*[58,59], or in *Aiptasia* where over 3000 TGRs are linked to endosymbiosis and other unique traits[8]. Similarly, in the *T. maxima* genome, enriched functions in the TRGs are mainly related to host-symbiont maintenance and metabolic exchange. These functions include anti-oxidation processes, immune system responses, carbon concentration mechanisms, and transmembrane transport systems (Table 1, Supplementary Data 9). A unique set of TRGs only found in *T. maxima* are related to ciliary structure and function. These include genes coding for Cilia and Flagella Associated Protein 61, Intraflagellar Transport Proteins 20 and 172, along with other genes involved in cilium assembly (Supplementary Data 7). A previous histological study confirmed that in *T. gigas*, cells interacting directly with symbionts in their tubule system have cilia[19]. In addition, comparative transcriptomics studies found that cilium-related genes are upregulated in normal symbiotic conditions compared to when symbiosis is disrupted by darkness[60]. This suggests that the cilium-related genes found exclusively in *T. maxima* may be linked to the unique photosymbiotic trait of this species.

A significant portion of the TRGs and expanded gene families remain functionally unannotated. This highlights a prevalent challenge in gene annotation in non-model organisms, where limited reference data impede the identification of unique genetic components[61]. These uncharacterized genes could be key to further understanding unique aspects of giant clam evolution and underscore the need for increased genomic research in non-model species.

The *T. maxima* genome exhibits one of the highest known repeat contents (68.07%) in Metazoa, with over 55% of the genome consisting of transposable elements (TEs). *T. crocea* and *T. gigas* also display comparable

high repeat content[62]. On the other hand, the common cockle *C. edule*, a non-symbiotic species of the same family, has a repeat content of only 37.81%. This suggests that elevated repeats are not a characteristic of Cardiidae more broadly, but a unique trait to giant clams. In fact, a high proportion of TEs in symbiotic genomes might be the norm rather than the exception. This pattern is observed in chemosymbiotic bivalves[11], zoantharians[63], anemones[8], stony corals[8], and *Symbiodinium*[64], which all exhibit expanded TEs compared to their non-symbiotic relatives. We propose the following non-mutually exclusive hypotheses which may explain this phenomenon.

**Table 2 | Comparison of repeat content across *Tridacna* species and *Cerastoderma edule***

| Species | C. edule | T. crocea | T. gigas | T. maxima |
|---|---|---|---|---|
| Genome size (MB) | 755 | 1048 | 1176 | 1320 |
| Repeat type | Percentage (%) | | | |
| LINE | 2.87 | 2.15 | 2.05 | 2.61 |
| SINE | 11.35 | 16.01 | 19.28 | 15.91 |
| LTR | 0.71 | 2.51 | 1.23 | 2.4 |
| DNA transposons | 7.44 | 14.19 | 15.85 | 15.02 |
| Rolling-circles | 2.67 | 22.48 | 20.55 | 23.26 |
| Small RNA | 3.16 | 3.13 | 3.06 | 3.18 |
| Satellites | 0.04 | 0.19 | 0.26 | 0.2 |
| Simple repeats | 1.12 | 0.91 | 1.05 | 0.89 |
| Low complexity | 0.15 | 0.13 | 0.16 | 0.13 |
| Unclassified | 11.34 | 9.87 | 6.8 | 7.5 |
| Masked Total | 37.81 | 68.58 | 67.43 | 68.07 |

Comparison of repeat content in *Tridacna maxima*, *Tridacna crocea*, *Tridacna gigas* and *Cerastoderma edule*, highlighting the significantly higher proportion of repeats, especially transposable elements (TEs) in *Tridacna*.

Symbiotic relationships create environments where genetic material may transfer between hosts and symbionts more easily. For example, symbiotic anemones and corals harbor horizontally transferred (HT) genes from their symbionts, some are involved in ultraviolet radiation photoprotection, crucial for optimizing photosynthetic efficiency[65]. Given the mobile nature of TEs, it is expected that HT of TEs can occur frequently[66–68], leading to rapid TE expansions after the origin of symbiosis, as observed in the SINE, LINE, and LTR categories. Future studies need to investigate whether host and symbiont genomes share more-than-average TE content.

Another hypothesized mechanism that explains TE expansions after photosymbiosis establishment is a suppressed immune system, which can be a result of the contracted immune gene families and immune pathway inhibitions by their Symbiodiniaceae symbiont[69]. Firstly, a suppressed immune system can be more susceptible to viral infections, which is a major source of TE insertion. Further, suppressed immune functions can lead to a reduction in TE silencing mechanisms, such as RNA interference[70] and DNA methylation[71]. Altogether, the host genome may be more vulnerable to TE integration and lack common mechanisms to remove them once integrated.

On the other hand, TE expansion may not be a side effect of symbiosis, but a crucial component of symbiosis adaptation through genome innovations. TEs introduce regulatory elements and genes into host genomes that can be co-opted for new host functions, leading to the evolution of novel genes and regulatory elements[72,73]. For example, in plants, TEs facilitate gene duplication, influence gene expression, and affect chromatin structure, thereby impacting the regulation of immune responses[74]. Indirectly, TEs trigger evolutionary arms race with host genomes, and various host-TE silencing mechanisms, such as the Krüppel-associated box zinc finger proteins (KRAB-ZFPs) that end up acquiring new functions beyond TE silencing, contributing raw material for genetic evolution[72]. The *T. maxima* genome showed expansion of numerous zinc finger gene families, which may indeed be an effect of TE-induced genomic innovation. This hypothesis is further strengthened by the observed expanded gene family duplication patterns. More than 90% of expanded genes in *Tridacna* were dispersedly

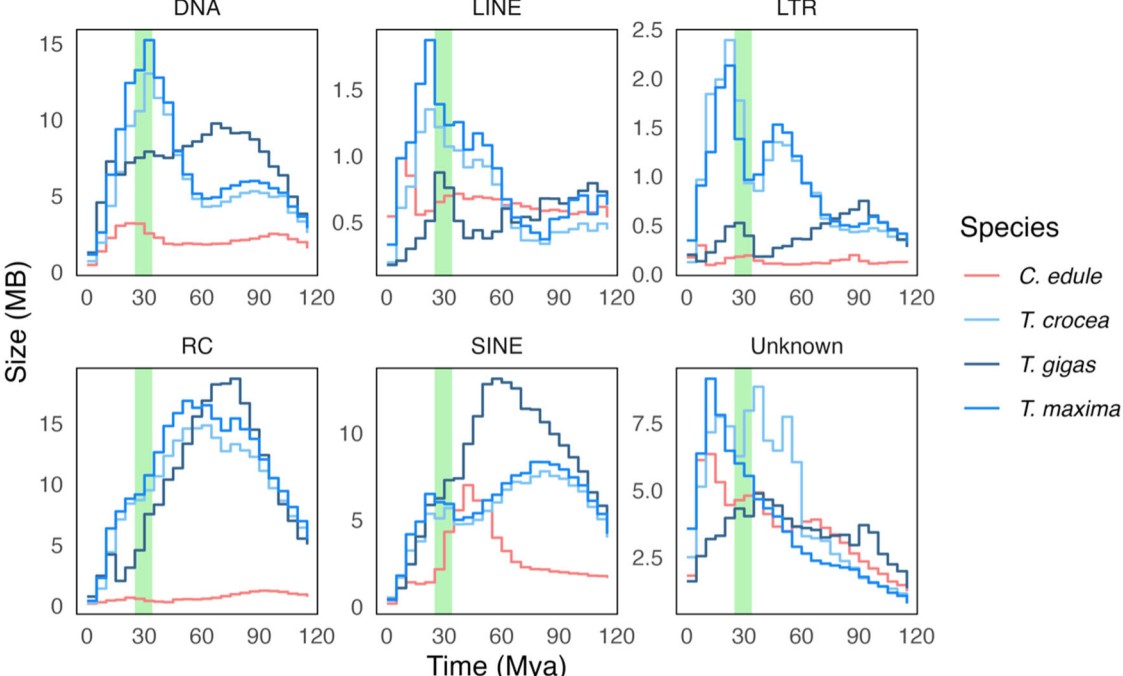

**Fig. 5 | Insertion time estimates for major transposable element families.** Estimation of insertion times for major families of transposable elements in *Tridacna maxima*, *Tridacna crocea*, *Tridacna gigas* and *Cerastoderma edule*, with the light green band representing the origin of stable photosymbiosis in giant clams during the Eocene-Oligocene transition period. DNA DNA transposons, RC Rolling-Circle transposons, LINE Long Interspersed Nuclear Elements, SINE Short Interspersed Nuclear Elements, LTR Long Terminal Repeats.

duplicated, which is a signature of TE-facilitated gene duplication[75], in contrast with the most common form of gene duplication induced by DNA replication errors.

Unlike SINE, LINE, and LTR, the TE families DNA and RC underwent rapid expansion long before the emergence of photosymbiosis. This indicates that additional mechanisms contributed to the elevated portion of these TEs, which are currently underexplored. Our understanding of TE function remains limited despite their impact on genomic regulation and evolution[76]. While numerous algorithms and pipelines exist for the automated identification of putative TE families, generating a library of high-quality TE consensus sequences typically requires manual curation[77], which is more challenging for non-model organisms. Investigating TEs in a broader range of organisms will be crucial for understanding the evolutionary reciprocal influences between TEs and symbiotic organisms.

## Conclusion

Symbiotic associations have played a pivotal role in shaping animal evolution, contributing to the vast diversity of ecological niches, morphology, and lifestyles across animal phyla[2], and leading to the ecological success of some of the most productive ecosystems like coral reefs and hydrothermal vents. Here, our exploration of a high-quality genome from the giant clam *Tridacna maxima* provided new insights into the genomic foundations that underlie the intricate photosymbiotic interactions between a host and its symbionts. We showed that symbiotic associations may impact animal genome evolution, at both gene and structural levels. We demonstrated that various aspects of the *T. maxima* genome such gene family and repetitive element evolution, could be influenced by their photosymbiosis ecology. Our comparative approach highlighted both similarities and differences to trends newly revealed in other symbiotic invertebrate genomes[8]. Genome-based inferences of *T. maxima* historical demography indicate that a tight correlation exists between global *T. maxima* population shifts and the expansion and decline of shallow marine reef habitats. This finding not only underscores the grave impact of present climate-related threats to IUCN Red-listed/CITES protected giant clams, but also emphasizes the urgency of conservation efforts to mitigate threats to coral reef ecosystems. Looking ahead, generating high-quality reference genomes from both hosts and symbionts will be essential to fully understand symbiotic relationships. Recent progress in sequencing technology and experimental manipulation provides opportunities to explore the genomic evolution, innovations, and constraints driven by these symbiotic associations. As an example, the work presented here is a major step towards advancing our understanding of the genomic drivers underlying metazoan evolution and diversification.

## Methods

### Sampling, genome sequencing and assembly

A live adult specimen of *Tridacna maxima* was collected from Luminao reef flat in Guam at 1 meter depth under collection permit SC-MPA-20-003, and transported to the University of Guam Marine Laboratory. Adductor muscle tissue samples were flash frozen in liquid nitrogen before being shipped for library construction and sequencing (Export permit: CO-21-010). DNA was extracted using the Qiagen mini kit (Qiagen, Hilden, Germany). Due to the presence of small DNA fragments in the samples, an unsheared DNA size selection was performed with a cutoff at 20 kb. The PacBio SMRTbell library for PacBio Sequel was constructed using SMRTbell Express Template Prep Kit 2.0 (PacBio, Menlo Park, CA, USA) using the manufacturer-recommended protocol. Whole genome sequencing was conducted by Dovetail Genomics (Scotts Valley, CA, USA) using a combination of long-read PacBio sequencing at ~60X depth on the PacBio Sequel II platform and Dovetail Hi-C Omni-C libraries sequenced on an Illumina HiSeq X Ten at ~30X depth. PacBio reads were de novo assembled with the WTDBG2 pipeline with default parameters. The resulting assembly was compared against the nt database to detect potential contamination with blobtools v1.1.1 to remove possible contaminations. Purge_dups v1.2.3 further refined the assembly by eliminating haplotigs and overlapping contigs. Then initial assembly was used as input for the Dovetail HiRise

Scaffolding pipeline[78], which used the proximity ligation data from the Omni-C library reads to identify putative breaks and joins and produce the final chromosome-level assembly.

### Transcriptome sequencing and genome annotation

Total RNA was extracted from the adductor muscle of the same *T. maxima* individual using a RNeasy kit (Qiagen, Hildenheim, Germany) on a QIAcube DNA/RNA extraction robot (Qiagen, Hildenheim, Germany) following manufacturers' protocol. cDNA libraries were prepared using an Illumina NeoPrep system (Illumina, San Diego, CA, USA), and paired-end sequenced (75 bp) on an Illumina NextSeq 500 at the University of Guam Marine Laboratory, Guam. RNAseq libraries were de novo assembled using Trinity v.2.9.1[79] after quality trimming using Trimmomatic v.0.35[80]. Two additional published transcriptomes[16,81] were merged with our new transcriptome to facilitate genome annotation.

RepeatModeler v.2.0.1[82] and RepeatMasker 4.1.0[82] were used with default parameters to identify and annotate the repetitive content of the genome, and to obtain a soft-masked version of the assembly. Gene models were predicted ab initio with MAKER v.3.01.03[83], using the transcriptome sequences to train Augustus v.3.3.3[84]. Subsequently, all *T. maxima* RNAseq reads available at the NCBI SRA database (48 libraries, including different tissue samples: muscle, mantle, visceral mass, kidney, gonads, gills and byssus, see Supplementary Data 12) were used to improve the ab initio genome annotation. RNAseq reads were downloaded using sratoolkit v.3.0.0 (https://github.com/ncbi/sra-tools/wiki/01.-Downloading-SRA-Toolkit) and mapped to the genome assembly using STAR v.2.7.4a[85]. Gene transfer files (GTFs) were generated from each individual bam file and subsequently merged using StringTie v.2.2.1[86]. Gene models from the RNAseq data were predicted using TransDecoder v.5.6.0 (https://github.com/TransDecoder/TransDecoder), with homology searches to annotate and retain open reading frames (ORFs) with functional significance. BLASTP v.2.10.0[87] and hmmscan v.3.3 (https://hmmer.org) were used for homology searches against Uniref90 and Pfam databases, respectively. TransDecoder and ab initio MAKER annotations were merged using StringTie to obtain the final genome annotation. Gene completeness was assessed using BUSCO v.5.4.2[88] and manually curated following Moggioli et al.[89] using blastp[89]. Gene density and repeat density were visualized using circos v0.69.9 (https://circos.ca/).

### Demographic history inference

Dynamic estimation of the effective population size ($N_e$) backwards in time was implemented using the Pairwise Sequential Markovian Coalescent (PSMC) method[25]. R1 and R2 Omni-C Illumina reads were mapped to the genome assembly independently using bwa mem v.0.7.17[90]. The resulting bam files were sorted by genome coordinate and merged with samtools v.1.10[91], and average depth was assessed with qualimap v.2.2.1[92]. Genotypes were called using samtools mpileup and bcftools call v.1.10.2 and subsequently filtered using the vcfutils script from the samtools/bcftools package with the following parameters: MinDepth = average depth/3, Max-Depth = average depth × 2. PSMC was applied with the following parameters (-N25 -t15 -r5 -p "4 + 25*2 + 4 + 6") to estimate historical $N_e$. A general bivalve mutation rate of $1.1 \times 10^{-8}$ per site per generation[23] calculated from 6 bivalves and a generation time of 10 years based on *T. maxima* reproductive cycle and longevity[93] were used for scaling the PSMC result to years.

### Gene family expansion, contraction, and taxonomically restricted genes

Annotated genes were mapped into gene families using OrthoDB v11[94] with default parameters. 29,695 gene families presented in 15 molluscan species were retrieved from OrthoDB v11. The annotated genes of *T. maxima* were assigned to those gene families. The sequences from those single copy gene families were aligned with mafft v7.508[95], then trimmed with TrimAL v1.4 in automated mode (http://trimal.cgenomics.org/). The trimmed alignments were concatenated to a super-matrix. RAxML v8.2.12[96] was used to

infer phylogenetic trees with 1000 bootstraps. Then the phylogeny was time calibrated using RelTime embedded in MEGAX v10.2.4[97] with a divergence time estimated utilizing the split of Bivalvia and Gastropoda at 529.8 mya. The cephalopod lineage was truncated due to its distant relation to *T. maxima*. With the gene count table and molluscan genomic phylogeny generated from the previous step, gene family expansion and extraction were predicted using CAFÉ v5[98]. A wide range of parameters were used, and the best model was selected by comparing likelihood values. Significant ($p < 0.05$) expanded and contracted families were selected by a custom bash script. Gene family annotations were retrieved from the orthoDB v11 database. Moreover, OrthoFinder with default parameters was used to retrieve taxonomically restricted genes (not present in other molluscan species) and gene families in *T. maxima*. Analyses on the three *Tridacna* species were conducted similarly to those performed on *T. maxima*, though the annotation of expanded gene families differed. Individual functional gene annotations were performed using emapper v5, then the most frequent annotation was applied to each gene family respectively. A more conservative strategy was employed when selecting expanded and contracted gene families. Only gene families shared by all three *Tridacna* species, with more than one copy and absent in the other 13 molluscan species, were selected. Gene families that were either missing or had only one copy in *Tridacna* but were widespread in the other 13 molluscan species were presented in Fig. 5.

### Gene duplication identification
Duplicated genes were identified with MCScanX[99] with default parameters using as input the BLASTP output file of all proteins against a blast database of all proteins (all-vs-all). The MCScanX script "duplicate_gene_classifier" was used to classify the duplicated genes in tandem, segmental, dispersal and segmental duplications. This classification was used to characterize the genes belonging to expanded families in *Tridacna* into each duplication category using custom bash scripts.

### Repetitive element annotation and transposable elements insertion time estimation
To further assess repeat elements in the *T. maxima* genome, a multiple-step method was used to annotate repeat elements. First, unknown elements in the de novo repeat library were annotated using RepeatMasker v4.1.0 with known elements for 5 iterative rounds. Then genome-wide repeats were identified and annotated with RepeatMasker v4.1.0 using four reference libraires sequentially: 1) simple repeats 2) Mollusca Repbase library v20181026 3) known repeats elements from the de novo library and 4) unknown repeats from the de novo library. Insertion time of major categories of repeat elements including transposable elements were estimated using parseRM.pl[100] with the same substitution rate as used in PSMC. To compare repeat elements composition with a non-symbiotic Cardiidae bivalve, the *C. edule* genome (GCA_947846245.1) was downloaded from NCBI GenBank and the same pipeline was applied.

### Synteny analysis among *Tridacna* genomes
Synteny among our newly sequenced *Tridacna maxima* genome and two other Tridacna species (*T. crocea* and *T. gigas*) was investigated using the GENESPACE v1.3.1 pipeline[101]. Custom bash scripts were used to format the bed files for GENESPACE. Briefly, AGAT v1.4.0 was used to extract protein sequences from the annotation files and to convert gff files to bed files, followed by formatting steps using awk, sed and cut bash commands. Riparian plots were generated by the GENESPACE R package.

### Reporting summary
Further information on research design is available in the Nature Portfolio Reporting Summary linked to this article.

### Data availability
The sequencing data generated for this study can be accessed through the National Center for Biotechnology Information (NCBI) Sequence Read Archive (SRA) under the BioProject PRJNA1077608. The genome assembly and annotation are deposited in NCBI Genome Portal under accession number GCA_045685785.1 (https://www.ncbi.nlm.nih.gov/datasets/genome/GCA_045685785.1/).

### Code availability
All scripts are openly available on the GitHub Repository: https://github.com/Ruiqi-CUB/TmaximaGiantClamGenome.

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

## Acknowledgements

This work is funded by a Packard Fellowship for Science and Engineering (2019–69653) and a Dovetail's Genome Assembly Award to J.L., and the National Science Foundation NSF-EPSCoR grant # OIA-1946352 to S.L. Publication of this article was funded by the University of Colorado Boulder Libraries Open Access Fund. We thank Constance Sartor for her contribution in preparing RNAseq libraries. We are indebted to Nathan J. Kenny for his invaluable help with genome annotation and to Aurora García-Berro for her help with the PSMC analysis.

## Author contributions

R.L., C.L., S.L., L.K. and J.L. designed the study. S.L. collected specimens and generated data. R.L. and C.L. carried out the experiments, generated data, and conducted analyses. All authors wrote the manuscript. The authors read and approved the final manuscript.

## Competing interests

The authors declare no competing interests.
