## [Transparent Peer Review file · Communications Biology]

Photosymbiosis Shaped Animal Genome Architecture and Gene Evolution as Revealed in Giant Clams

Corresponding Author: Dr Ruiqi Li

Version 0:

Reviewer comments:

Reviewer #1

(Remarks to the Author)

This manuscript presents a high quality genome for the photosymbiotic giant clam *Tridacna maxima*, and performs interesting analysis of demographic history. I believe that these elements are worthy of publication. Other analyses are flawed and many conclusions are overstated, therefore I believe the manuscript requires major revision before being accepted.

Main comments:

The authors state that 'Our genomic study provides a comprehensive assessment of how its photosymbiosis lifestyle influenced its evolution at multiple levels'. None of the analyses performed within the manuscript, except possibly the repetitive/transposable element analysis, was conducted in a way that would distinguish the influence of photosymbiosis, specifically, on genome evolution.

- The tandem gene duplication analysis does not determine when the genes were duplicated. These could be species-specific, genus-specific, or much older. Therefore very few inferences can be made through GO-term enrichment of duplicated genes and evolution of giant clam-specific features.
- The gene family analysis (and TRG analysis) suffers from a similar issue. The resolution of this analysis will depend on the species used in the analysis – which is not reported in the paper – although appears to utilize only the molluscs found within OrthoDB (mainly gastropods, or pteriomorphs). To perform a targeted analysis of which changes may have arisen due to photosymbiosis, this analysis should combine analysis of photosymbiotic *Tridacna* (for which there are now several genomes available) with closely-related non-photosymbiotic taxa.

Given these issues focus should be taken off the 'comprehensive assessment of how its photosymbiosis lifestyle influenced its evolution'. Statements such as 'Here, our meticulous exploration of a high-quality genome from the giant clam *Tridacna maxima* provided unprecedented new insights into the genomic foundations that underlie the intricate photosymbiotic interactions between a host and its symbionts' should be removed.

I am not at all convinced by the 'loss of Cdx'. What evidence is there to suggest that this isn't simply due to misannotation or incomplete genome sequencing? A quick blast of *Mytilus galloprovincialis* CDX (VDI62576.1) against translated *Tridacna* genomes reveals hits in *T. gigas* and other species; a quick check suggests that this may, in fact, be Cdx. From this, I think that loss in *T. maxima* is highly unlikely. To make this claim you would need very careful analysis of syntenic regions, perhaps a demonstration of failure of degenerate Cdx primers on *T. maxima* DNA (with appropriate positive controls), etc. You should also demonstrate that it can't be found in the other available *Tridacna* genomes.

It also seems odd as to why no comparative analysis is performed with other giant clam genomes, especially as one of these (*Tridacna gigas*) was sequenced by authors of this paper? *Tridacna gigas* was found to have 17 chromosomes, whereas *T. maxima* were here found to have 18 (and *T. crocea* was also reported to have 18). Is this a true difference and, if so, what is happening in terms of chromosomal evolution?

In general, I find the discussion far too speculative, and suggest the authors condense it down to the points for which there is the best evidence.

It would be good to include some more information in the introduction about the current state of knowledge of the evolution of giant clam photosymbiosis. Is this thought to be a single evolutionary event, or has it evolved in multiple giant clam lineages independently? When is it believed that this symbiosis evolved?

Other comments:

Lines 20-21: 'how symbiosis left complex signatures in an animal's genome'. The past tense in this sentence implies that the clam is no longer photosynthetic?

Line 26: Why do you say there is a novel arrangement of Hox genes?

Line 36: missing a full stop

Lines 44-45: "Growing work started to uncover" – awkward wording, I think due to tense. "Growing work has started to uncover" sounds better.

Lines 46-46: Simply listing genes here isn't particularly informative. Can you briefly outline the results of these studies (what is the influence of symbiosis on these genes? Or, what are the molecular mechanisms that you mention?).

Lines 51-53: What exactly is intriguing about having a higher proportion of transposable elements? Statement needs fleshing out.

Line 98: Start new sentence for "86% of the genome..."

Lines 106-108: Manual annotation to improve BUSCO scores is somewhat misleading, and it is unclear exactly how this was done. Suggest including a sentence to say something like: 'This indicated that the genome coverage is high, but that some gene annotations are lacking'. Additional detail needs to be added to methods.

Line 223 onward: "our BUSCO analysis suggested faster rates of molecular evolution in the coding sequences of *T. maxima*". How have you come to this conclusion? I can't see anything about this in the results?

Line 233 onward: "our analyses pinpointed to the genomic basis of giant clams' large shell size, and unique regulation of the calcification processes". The basis for this statement is tandem duplication of genes and gene family expansions, and is problematic because 1) the tandem duplications are not dated, and therefore could pre-date evolution of the large shell, and 2) all of these genes and GO terms are multifunctional (none are shell-specific), so their expansion could quite easily be due to other functions.

Lines 399-400: More details required here; how was DNA extraction performed (this is notoriously tricky for molluscan tissue?). Was any size selection/inhibitor removal performed?

Figure 1 is not referred to in the text. '*Tridacna maxima*' is misspelt in the figure.

Figure 2 – the text on the outside of the circle is unreadable at this size. Suggest reducing the size (width) of panel B and increasing A. Also, it's quite hard to relate the x-axis to the dates given in the text (e.g., 3mya) given the log scale. Suggest making this easier – e.g., indicating important timepoints specifically.

Reviewer #2

(Remarks to the Author)

Review of Manuscript COMMSBIO-24-1281-T: "Photosymbiosis Shaped Animal Genome Architecture and Gene Evolution as Revealed in Giant Clams":

This manuscript by Li et al presents a new giant clam genome (*Tridacna maxima*), which the authors use to investigate the signatures that symbiosis creates in a genome. The genome itself appears to be high quality and will be incredibly useful for investigating photosymbiosis in giant clams and the transposable element connection to symbiosis is extremely interesting.

However, my primary difficulty with this manuscript is the interpretation of the results. I appreciate that the focus is on symbiosis, but this is one giant clam genome compared to just a few other non-symbiotic taxa. I find the stories included in the discussion to be unsubstantiated in many cases (as highlighted in my comments below). I understand the focus on broad gene expansion and contraction, but there are a lot of assumptions based on blast annotation in the interpretation of the results, without any functional data to back them up.

Second, the demographic history section in the results came out of nowhere for me. How do demographic history changes tell us more about how symbiosis impacts the genome? The discussion goes into this a bit more and I can kind of see the logic, but the connection to the genome signatures story is not very strong.

Finally, some of the methodological procedures need to be explained more sufficiently (more below).

Introduction

Page 2, Lines 51-53: Why is this intriguing? I'm not following the connection here. And the use of the word limited is confusing. Does this mean there is some evidence that shows the opposite?

Page 2, Lines 55-57: What evidence exists that their large size is due to photosymbiosis? There are certainly large animals without symbiosis and small animals with symbiosis. A clearer explanation would be helpful here.

Results

Page 3, Lines 106-108: This manual curation of only BUSCOs appears to be an artificial inflation of the BUSCO score given that non-BUSCOs would not change during this process. Perhaps clearer explanation of the goal of this would be helpful, as well as a bit of text explaining why those were not annotated correctly in the first place.

Discussion

Page 5, Line 223: Faster rates compared to what?

Page 5, Lines 224-225: And yet, the BUSCO analysis found >90% of these genes without manual curation, which is pretty consistent to other mollusk genomes. If the rates of evolution are that fast, why were you even able to manually curate them?

Page 5, Lines 226-227: Is it possible that Cdx was just not picked up in the analysis? If you had to manually annotate BUSCO genes, it seems like it would be possible that Cdx might not be annotated through automated pipelines. If this was checked manually, that information should be added.

Page 6, Lines 230-231: How would a loss of developmental regulation lead to specialization? Please explain.

Page 6, Lines 235-236: But chitin binding and calcium ion binding are not exclusive to shell matrix proteins. What else could these be relevant to in the giant clam?

Page 6, Lines 239-242: How exactly are the authors imagining that expansions and tandem repeats of genes may be responsible for shell size? Is this something that has been seen in other organisms? The expression patterns of these genes have not been identified; how do they even know they are used for biomineralization at all in this species? It also isn't clear how these functions relate to symbiosis, which is meant to be the primary focus of the manuscript.

Page 6, Lines 244-Page 7, Line 279: These whole paragraphs feel very hand-wavy to me. Is there any evidence of a connection between these genes and symbiosis? The example given is in a mouse, which does not have symbionts, and the rest of the argument seems to me to just say that symbionts provide energy (which we have evidence for) and that means that they have enough energy to get big so they don't need as many CTRP genes. But again, the authors don't compare with any other symbiotic organisms, so I have no idea whether this is just specific to this giant clam species. The same feels true of the immune-related genes in the next paragraph.

Page 7, Lines 291-292: Long compared to what? And I would argue this is a hypothesis based on these data and prior information, not support of an assumption.

Page 7, Lines 296-297: What % and how similar?

Page 7, Lines 306-308: This sentence comes out of left field for me. How does this relate to expanded gene families? There seem to be steps missing between nutrient benefits of photosymbiosis, positive selection, and gene family expansion. The connections are not clear to me based off of this text.

Page 8, Lines 335-341: I am confused by this paragraph. Suppression of immune systems and inhibition of host immune pathways are individual interactions. How are these interactions expected to lead to TE expansions across evolution?

Methods

Page 9, Line 403 (and beyond): I'll use WTDBG2 as an example, but this goes for programs used across the pipeline; what parameters were used? If it was just defaults for everything, please provide that information. If not, the specific parameters used should be listed somewhere either in the main text or a supplement.

Page 10, Lines 418-419: Were these two transcriptomes merged with the generated assembly or used separately? Please clarify.

Page 10, Lines 424-427: Why were these not included in the initial Augustus training? It would likely have been helpful to include as much RNAseq data as possible.

Page 10, Lines 430-434: Were these compared to gene models from MAKER? It isn't clear to me why these steps were only performed on a subset of gene models from the RNAseq data but not on all gene models.

Page 10, Lines 434-435: How were duplicates in the annotation handled when merging gene prediction annotations?

Page 10, Lines 456-457: But which proteins? From both sets of gene models (MAKER and RNAseq)?

Page 11, Line 460: Why the focus on tandem duplications? There are likely many other types of duplications present in the genome, but these would not be accounted for in this analysis unless I'm mistaken. Is there a specific reason for this choice?

Page 11, Line 480: An explanation of the metric used to determine significance is necessary here.

Page 11, Lines 481-483: How exactly was this done? Were the same proteomes used? With what parameters? At what levels were the taxonomically restricted genes isolated? Just the species level? Also, it certainly would be possible that OrthoFinder would result in different gene families or clusters than OrthoDB, so it would be helpful to describe how gene families were confirmed across the two pipelines in terms of their membership.

Figures

Figure 1: I don't find this figure very helpful. What is the main point I am supposed to take from it? The authors do not even investigate morphological adaptations or the immune system really, and one could argue that coding regions are divergent in every species.

Figure 6: This figure suggests that many of the major expansions in *Tridacna* likely emerged prior to the origin of stable photosymbiosis. I also just realized that the authors use size for this rather than percentage of the genome. Given that the genome is just larger, would it not be useful to provide information on both? For example, in the table it is obvious that *Tridacna* has a much higher percentage of DNA Transposons and Rolling-Circle Transposons compared to *Cerastoderma*. It would be interesting to know at what point in time those differences emerged.

Version 1:

Reviewer comments:

Reviewer #1

(Remarks to the Author)

The authors have made corrections that have significantly improved the manuscript (especially the synteny analysis and comparison to other *Tridacna* genomes, which are both very interesting). I believe some corrections are still required before the manuscript is acceptable for publication.

I find that the claims in the abstract and conclusion are still too overstated. The authors pointed out in their rebuttal that 'in order to definitively link photosymbiosis to any genomic features, one needs to either conduct genetic manipulations on the organism to see how they disrupt photosymbiosis, or use multiple symbiotic/non-symbiotic sister groups to test if certain genomic features are consistently associated with the symbiotic lineage. The former method is very challenging for long-lived non-model study systems like giant clams'. I completely understand this, however this does not remove the need for caveats and caution in interpretation. For example, in the abstract:

Line lines 24-25: 'highlighting how abiotic and biotic factors dictate *T. maxima* microevolution' needs to be rephrased to "revealing how abiotic and biotic factors may dictate *T. maxima* microevolution"

Line 25: "Comparative analyses revealed unique symbiosis-driven genomic features" needs to be rephrased to "Comparative analyses revealed genomic features that may be symbiosis-driven".

Similar corrections need to be made to many sentences in the conclusion, too.

The analysis of 'taxonomically restricted genes' is problematic, as it appears to have been conducted through a comparison of *T. maxima* with other non-*Tridacna* genomes. This is odd, as other parts of the manuscript include the two other *Tridacna* genomes. If the main question is about potential links to photosymbiosis, why wouldn't you just compare *Tridacna*-specific genes to other non-photosymbiotic genomes, rather than the 2-step analysis conducted here? This creates issues in the discussion, for example, the statement 'A unique set of TRGs only found in *T. maxima* are related to ciliary structure and function' implies that these genes are not found in the other *Tridacna* genomes, which is not really what you mean. In my opinion it would be best if this analysis were redone to include all three *Tridacna* genomes, however I understand that this is a significant amount of additional work. Otherwise, careful explanation and rephrasing of the relevant sections is required to resolve the issue.

Other comments:

Line 52: change to 'emerging evidence shows'

Line 62: change to 'is thought to have'

Line 81-82: 'VHA' is uninformative, change to 'V-type H⁺-ATPase'

Lines 151-152: Unclear here whether the 'other mollusc genomes' includes the other *Tridacna* genomes, so are you discussing genes and gene families likely unique to *T. maxima*, or all *Tridacna*? Reading below it's clear that you are not including the other *Tridacna* genomes, but this seems a strange way to go about the analysis (see comment above)

Line 186: Change 'known bivalve substitution rates' to 'calculated' or 'predicted' bivalve substitution rates. Also, would you expect these to be the same in TE sequences, or elevated in comparison with the rest of the genome?

Line 189: I'm assuming this should be a reference to Fig. 5.

Line 205-208: The wording is too strong here – given that photosymbiosis was thought to have originated in stem *Tridacninae* at ~27mya, and that your demographic history inference only extends back to 3mya, you can't link the acquisition of photosymbiotic traits to anything, really.

Line 208-216: Again, this is all too strongly worded. Your analysis is a model/estimate only, and you cannot definitively link historical climatic events to the (estimated) population size, you can only say that your estimates are congruent with these hypotheses.

Line 221: change 'must have' to 'likely'

Line 224: change to 'is estimated to have occurred around the same time'

Lines 225-226: change to 'likely also made their survival'
Line 228: replace 'observed' with 'estimated'

Lines 236-240: this small paragraph on genes potentially related to shell evolution is out-of-the-blue, given shells are not mentioned at all prior to this (likely needs some description of shell evolution in the introduction). It also highlights the major difficulty with this analysis; photosymbiosis is not the only trait that has evolved in this lineage, and gene family expansion/contraction can be associated with any of these traits. It's fine to speculate, but the language needs to reflect the uncertainty throughout.

Lines 242-243: I can't see anything related to this gene family in the results section – it's present in Fig 4, but if this is 'striking' it should be explicitly pointed out in the results text.

Line 249: change to 'has been selected against'

Lines 250-251: Delete the 'and' at the beginning of the sentence.

Line 262: change 'highlights' to 'may reflect the'

Lines 269-279: the text here is overly long, repetitive, and entirely speculative. Rewrite to make more concise.

Lines 336-335: how is an increased susceptibility to viral infections relevant to the discussion of TE content?

Lines 336-337: how is symbiont suppression of host immune pathways relevant? Or is this an explanation of the immune system suppression mentioned in the first sentence of the paragraph? This paragraph needs restructuring to make the points more clearly.

Lines 348-349: Given the preceding sentence wouldn't you expect expansion of KRAB-ZFPs only? Argument needs further development.

Lines 355-356: Sentence needs revising to something like 'Unlike (XYZ TEs), the DNA and RC families likely underwent...'

Lines 360-361: Why is this extremely challenging for non-model organisms? It doesn't require functional techniques, so should be no more difficult in non-model organisms than in model organisms

Line 370: delete 'meticulous'

Figure 1: 'expansion and contraction of metabolic genes', change to either 'gene families' or 'gene repertoire' (the genes themselves are not expanded or contracted)

Reviewer #2

(Remarks to the Author)

This manuscript by Li et al presents a new giant clam genome (*Tridacna maxima*), which the authors use to investigate the signatures that symbiosis creates in a genome. The genome is of high quality and will be incredibly useful for investigating photosymbiosis in giant clams, and I find the addition of the two other *Tridacna* genomes more compelling. Overall, I find the manuscript much improved. I have just a couple additional minor points below:

Page 6, Lines 242-251: Has this gene (or others like it) been found to be relevant to symbiosis in other organisms, like corals? It would be helpful to have some additional context here about other symbiosis systems.

Page 7, Lines 301-303: Does that mean that the authors would expect the other *Tridacna* species to use different systems? It seems like a big deal that these genes were not found in the other *Tridacna* species. Or do the authors mean that these genes were not found in other mollusks? I'm a bit confused.

Version 2:

Reviewer comments:

Reviewer #1

(Remarks to the Author)

I believe the authors have satisfactorily addressed all highlighted concerns.

Reviewer #2

(Remarks to the Author)

The manuscript by Li et al is much improved from previous versions, and the authors have modified their conclusions in way that appears more realistic. I recommend acceptance.

Reviewer #1 (Remarks to the Author):

This manuscript presents a high quality genome for the photosymbiotic giant clam *Tridacna maxima*, and performs interesting analysis of demographic history. I believe that these elements are worthy of publication. Other analyses are flawed and many conclusions are overstated, therefore I believe the manuscript requires major revision before being accepted.

Response: We thank the reviewer for the detailed and constructive comments. As you will see in the following sections, we have included more analyses and addressed all the concerns. We believe the revisions have greatly strengthened the manuscript and allowed us to gain further insights of the data.

Main comments:

The authors state that ‘Our genomic study provides a comprehensive assessment of how its photosymbiosis lifestyle influenced its evolution at multiple levels’. None of the analyses performed within the manuscript, except possibly the repetitive/transposable element analysis, was conducted in a way that would distinguish the influence of photosymbiosis, specifically, on genome evolution.

Response: We understand and agree with the reviewer’s concerns on whether the genomic characters discussed in this manuscript can be causally linked to the photosymbiosis ecology. From our understanding, in order to definitively link photosymbiosis to any genomic features, one needs to either conduct genetic manipulations on the organism to see how they disrupt photosymbiosis, or use multiple symbiotic/non-symbiotic sister groups to test if certain genomic features are consistently associated with the symbiotic lineage. The former method is very challenging for long-lived non-model study systems like giant clams. Even if it is possible, it will still require the pre-identification of a set of candidate genetic components, which this current manuscript is trying to provide.

The second approach is only possible if photosymbiosis independently evolved multiple times in a clade, otherwise it is difficult to tease apart whether a novel genomic feature is the result of photosymbiosis or is just a lineage specific feature. In the giant clams, photosymbiosis only evolved once in their common ancestor, so we unfortunately cannot do multi-pair comparisons. What we can do, is to compare *Tridacna* genomes with non-symbiotic cardiid genomes, and hypothesize that some of the unique features (such as expanded gene families and transposable elements) are results of photosymbiosis. We then look for supporting evidence from other photosymbiotic systems (such as corals and anemones) in the literature to support our hypothesis.

In the revised manuscript, we have included two additional genomes from other *Tridacna* species, *Tridacna crocea* that is closely related to *Tridacna maxima*, and *Tridacna gigas*, which is more

distantly related. We show that the proposed features such as the unique immune system and high TE load, exist in the other *Tridacna* as well. We also discussed that these features have been identified in other photosymbiotic animals. Again, this cannot be the definitive test using a genomic approach due to the nature of how photosymbiosis evolved in *Tridacna*, but this is the best evidence we can obtain currently, and it lays foundation for future functional studies. We have also removed some analyses which provided weak arguments related to photosymbiosis, as the reviewer pointed out. Please see our detailed responses below.

The tandem gene duplication analysis does not determine when the genes were duplicated. These could be species-specific, genus-specific, or much older. Therefore very few inferences can be made through GO-term enrichment of duplicated genes and evolution of giant clam-specific features.

Response: We agree with reviewer's comment. Indeed, we can't know when these tandem duplications happened. Hence, following this comment, we removed this particular analysis from the revised version of the manuscript.

In the new version, we conducted a different duplication analysis to characterize the expanded gene families in the CAFE results (which are phylogenetically aware). We investigated why these gene families were expanded in *Tridacna* by assigning them into four categories of gene duplication: tandem, proximal, segmental and dispersed. We found that most genes found in the *Tridacna* specific expanded families were dispersedly duplicated (1,032 genes, 91.1%). Interestingly, literature on gene duplication indicated that “duplicates generated by transposable elements (TEs) can act as pseudo-TEs and amplify to multiple copies via transposition. The distinct copies are often dispersed on different chromosomes, which stands in contrast with the most frequent form of gene duplication induced by errors during DNA replication.” (Ma et al., 2023). This results further support our hypothesis that the expanded TEs in *Tridacna* genomes may play a role in driving genomic adaptations to *Tridacna* photosymbiosis.

Ma, H., Wang, M., Zhang, Y.E. and Tan, S., 2023. The power of “controllers”: Transposon-mediated duplicated genes evolve towards neofunctionalization. *Journal of Genetics and Genomics*, 50(7), pp.462-472.

Revisions:

Line 164-170: “Gene duplication analyses were conducted to characterize the nature of duplication for genes found in the expanded families. For *T. maxima*, most genes were dispersedly duplicated (3371 genes, 88.3%), followed by 248 (6.5%) genes in proximal duplications, 137 (3.6%) in tandem duplications, and 61 (1.6%) in segmental duplications. Similar patterns were found in expanded gene families shared by all three *Tridacna* species: 1,032 genes (91.1%) were dispersedly duplicated, 38 genes (3.4%) in proximal duplications, 28 (2.5%) in segmental duplications, and 25 (2.2%) tandemly duplicated.”

Line 350-353: More than 90% of expanded genes in *Tridacna* were dispersedly duplicated, which is a signature of TE-facilitated gene duplication⁷⁵, in contrast with the most common form of gene duplication induced by DNA replication errors.

The gene family analysis (and TRG analysis) suffers from a similar issue. The resolution of this analysis will depend on the species used in the analysis – which is not reported in the paper – although appears to utilize only the molluscs found within OrthoDB (mainly gastropods, or pteriomorphs). To perform a targeted analysis of which changes may have arisen due to photosymbiosis, this analysis should combine analysis of photosymbiotic *Tridacna* (for which there are now several genomes available) with closely-related non-photosymbiotic taxa. Given these issues focus should be taken off the ‘comprehensive assessment of how its photosymbiosis lifestyle influenced its evolution’. Statements such as ‘Here, our meticulous exploration of a high-quality genome from the giant clam *Tridacna maxima* provided unprecedented new insights into the genomic foundations that underlie the intricate photosymbiotic interactions between a host and its symbionts’ should be removed.

Response: We appreciate the reviewer’s insightful comments regarding the gene family and TRG analyses. We acknowledge that ideally, a comparison with non-photosymbiotic taxa closely related to giant clams would provide deeper insights. However, we are limited by the availability of genomic resources. At present, there is only one genome (*Cerastoderma edule*) available for non-symbiotic cardiid species, which are phylogenetically closest to *Tridacna*.

When selecting genomes for comparative analyses, we took both phylogenetic relationships and genome annotation quality into account. For example, although *Cerastoderma edule* is closely related to giant clams, its genome is poorly annotated (therefore many genes are unidentified). Thus, it can only be included in the repetitive element analyses (which do not require information on gene identity), but not in the gene family expansion/contraction analyses. The genomes from OrthoDB were chosen not only because of their high quality but also due to the comprehensive annotation of gene families documented in OrthoDB, which provide reliable annotations of contracted gene families.

In addition to our original analyses, we now provided new gene family expansion/contraction and TRG analyses that include two additional *Tridacna* genomes (*T. crocea*, *T. gigas*), and we approached the interpretation with caution. In addition to checking the p-values from the CAFE5 outputs, we selected gene families that are highly likely to be lineage-specific to giant clams, which could potentially be related to photosymbiosis. For instance, when presenting contracted gene families, we focused on those absent or represented by only one copy in *Tridacna* species but widespread and multi-copy in other molluscs. For expanded gene families, we selected those that are only present in *Tridacna* species, suggesting a possible lineage-specific function that may be related to photosymbiosis. We found that the originally described expanded and contracted gene functions (immune, metabolism, etc.) are consistent with the new analyses, after adding two additional *Tridacna* genomes.

In response to the reviewer's suggestion, we have adjusted our wording to reflect a more evidence-based interpretation of the results, focusing on the potential but non-definitive role of gene family changes in photosymbiosis:

Revisions:

Line 239-240: "Expansions of these biomineralization gene families may have driven diversification of gene functions and novelties responsible for giant clams' extraordinary growth rates and shell sizes⁴⁷."

Line 242-244: "The striking contraction of the C1q/Tumor Necrosis Factor-Related Protein (CTRP) gene family in *T. maxima* (and in the other two *Tridacna* species) may also be linked to its symbiont-fueled growth and ultimately large size."

Line 272-274: "This suppression likely facilitates the tolerance and long-term maintenance of symbionts by minimizing the host's immune reactions against them."

Line 299-303: "In addition, comparative transcriptomics studies found that cilium-related genes are upregulated in normal symbiotic conditions compared to when symbiosis is disrupted by darkness⁶⁰. This suggests that assumption that the cilium-related genes found exclusively in *T. maxima* may be linked to the unique photosymbiotic trait of this species."

I am not at all convinced by the 'loss of Cdx'. What evidence is there to suggest that this isn't simply due to misannotation or incomplete genome sequencing? A quick blast of *Mytilus galloprovincialis* CDX (VDI62576.1) against translated *Tridacna* genomes reveals hits in *T. gigas* and other species; a quick check suggests that this may, in fact, be Cdx. From this, I think that loss in *T. maxima* is highly unlikely. To make this claim you would need very careful analysis of syntenic regions, perhaps a demonstration of failure of degenerate Cdx primers on *T. maxima* DNA (with appropriate positive controls), etc. You should also demonstrate that it can't be found in the other available *Tridacna* genomes.

Response: This is a great point. Following reviewer's comment, we checked for the Cdx gene in other *Tridacna* genomes. We found the Cdx gene in both the *T. crocea* and *T. gigas* genomes and used their gene sequences to find the un-annotated Cdx gene in our *T. maxima* genome. It is unclear why Cdx was not detected in *T. maxima* previously after extensive effort. It is possible that *T. maxima* Cdx is very divergent and required sequences from congeners to be matched. The annotation was added to the annotation file and all relevant text was removed from the manuscript's new version.

It also seems odd as to why no comparative analysis is performed with other giant clam genomes, especially as one of these (*Tridacna gigas*) was sequenced by authors of this paper?

Response: Thank you for the suggestion. When this manuscript was written, many of the other *Tridacna* genomes were not available yet. Our group has since pushed forward with a few more genomes and published genome notes. Although the focus of this manuscript is still on *Tridacna*

maxima, we agree that comparative analyses with the other species are valuable. We therefore included *Tridacna crocea* and *Tridacna gigas* in several of the comparative analyses (chromosome comparison, gene family contraction and expansion, TE analyses) as stated above.

Tridacna gigas was found to have 17 chromosomes, whereas T. maxima were here found to have 18 (and T. crocea was also reported to have 18). Is this a true difference and, if so, what is happening in terms of chromosomal evolution?

Response: Following Reviewer's comment, we studied the chromosomal evolution in three *Tridacna* genomes: *T. maxima*, *T. crocea* and *T. gigas*. We performed a synteny analysis among the three genomes, added a new figure, and mentioned the chromosomal rearrangements that occurred in *Tridacna*.

Revisions:

Line 109-111: "However, *T. gigas*, a more distantly related congener, has one fewer chromosome, suggesting a possible chromosomal fusion or splitting event within the genus (Fig. 3)."

In general, I find the discussion far too speculative, and suggest the authors condense it down to the points for which there is the best evidence.

Response: We appreciate the reviewer's concern about the speculative nature of some parts of our discussion. We acknowledge that the ecological consequences we discussed are inferred from genomic data alone, and we agree that drawing conclusions without functional data is not ideal. However, due to the challenges of performing experimental manipulations on long-lived, non-model organisms like giant clams, we are currently constrained to genomic analyses. As one of the first genomes available for this lineage, our work is an initial step that we hope will inspire further functional studies.

Regarding the discussion points, particularly those about transposable elements, we recognize that these are speculative. We were careful to frame them as hypotheses, acknowledging that even in well-studied model organisms like *Drosophila* and mice, repetitive elements are not yet fully understood. That said, we did find concrete evidence of repetitive element expansion in giant clams, which mirrors similar expansions observed in other symbiotic organisms, such as corals, sea anemones, chemosymbiotic bivalves, and diatoms. We believe this parallel is unlikely to be coincidental, though we do not claim to have a full understanding of the mechanisms. Instead, we aim to propose hypotheses grounded in existing literature that can guide future research.

In light of the reviewer's suggestion, we have revised the wording of certain discussion points to ensure they are based on current evidence. We have made it explicit when we are proposing hypotheses.

Revisions:

Line 239-240: “Expansions of these biomineralization gene families may have driven diversification of gene functions and novelties responsible for giant clams’ extraordinary growth rates and shell sizes⁴⁷.”

Line 242-244: “The striking contraction of the C1q/Tumor Necrosis Factor-Related Protein (CTRP) gene family in *T. maxima* (and in the other two *Tridacna* species) may also be linked to its symbiont-fueled growth and ultimately large size.”

Line 272-274: “This suppression likely facilitates the tolerance and long-term maintenance of symbionts by minimizing the host's immune reactions against them.”

Line 299-303: “In addition, comparative transcriptomics studies found that cilium-related genes are upregulated in normal symbiotic conditions compared to when symbiosis is disrupted by darkness⁶⁰. This suggests that assumption that the cilium-related genes found exclusively in *T. maxima* may be linked to the unique photosymbiotic trait of this species.”

Line 321-322: “We propose the following non-mutually exclusive hypotheses which may explain this phenomenon.”

Line 332-334: “Another hypothesized mechanism that explains TE expansions after photosymbiosis establishment is a suppressed immune system, which can lead to a reduction in TE silencing mechanisms, such as RNA interference⁶⁹ and DNA methylation⁷⁰.”

Line 340-343: “On the other hand, TE expansion may not be a side effect of symbiosis, but a crucial component of symbiosis adaptation through genome innovations.”

It would be good to include some more information in the introduction about the **current state of knowledge of the evolution of giant clam photosymbiosis**. Is this thought to be a single evolutionary event, or has it evolved in multiple giant clam lineages independently? When is it believed that this symbiosis evolved?

Response: Thank you for inquiring about the information. These questions have indeed been answered in literature. All giant clams are photosymbiotic, and this relationship is thought to evolved once in their common ancestor at ~ 27 Ma (SD = 4.4) (Li et al., 2020).

Li, J., Lemer, S., Kirkendale, L., Bieler, R., Cavanaugh, C., & Giribet, G. (2020). Shedding light: a phylotranscriptomic perspective illuminates the origin of photosymbiosis in marine bivalves. *BMC evolutionary biology*, 20, 1-15.

Revisions:

Line 61-64: “Genetic evidence suggests that all giant clams are photosymbiotic, and this relationship is thought to evolved once in their common ancestor at ~ 27 mya (SD = 4.4)¹⁶, coinciding with the global expansion of modern coral reefs¹⁷.”

Other comments:

Lines 20-21: ‘how symbiosis left complex signatures in an animal’s genome’. The past tense in this sentence implies that the clam is no longer photosynthetic?

Response: Thank you for pointing this out. We have revised the sentence to use the present perfect tense, changing “left” to “has left”, to clarify that the clam remains photosymbiotic and that we are discussing ongoing genomic signatures of this relationship.

Revisions:

Line 19-21: “Here, we used a giant clam (*Tridacna maxima*) genome to demonstrate how symbiosis has left complex signatures in an animal’s genome.”

Line 26: Why do you say there is a novel arrangement of Hox genes?

Response: See response for above. In the new version of the manuscript, we have removed the How/Parahox section.

Line 36: missing a full stop

Response: Thank you for noticing this. We have added the missing full stop.

Lines 44-45: “Growing work started to uncover” – awkward wording, I think due to tense. “Growing work has started to uncover” sounds better.

Response: Thank you for the suggestion. We have revised the wording to “Growing work has started to uncover”

Revisions:

Line 43-44: “Growing work has started to uncover molecular mechanisms behind symbiosis through the lens of gene evolution.”

Lines 46-46: Simply listing genes here isn’t particularly informative. Can you briefly outline the results of these studies (what is the influence of symbiosis on these genes? Or, what are the molecular mechanisms that you mention?).

Response: We have revised the sentence to provide more context about symbiosis:

Revisions:

Line 45-47: “Such genes include putative pattern-recognition receptors involved in symbiont recognition, as well as transporters for carbon, nitrogen, phosphorus, and trace metals that facilitate host-symbiont metabolic exchange⁷⁻⁹.”

Lines 51-53: What exactly is intriguing about having a higher proportion of transposable elements? Statement needs fleshing out.

Response: Thank you for your comment. We find the higher proportion of transposable elements intriguing because traditionally, studies have focused primarily on protein-coding regions, often overlooking the potential significance of repetitive elements. We revised the sentence to highlight that this underexplored aspect of genome architecture may play a significant role in maintaining symbiotic relationships or be influenced by symbiosis:

Revisions:

Line 52-55: “Intriguingly, emerging evidence show that photosymbiotic animal genomes appear to possess higher proportions of transposable elements compared to non-symbiotic relatives^{8,11}, suggesting that this underexplored aspect of genome architecture may play a significant role in maintaining symbiotic relationships or be influenced by symbiosis.”

Line 98: Start new sentence for “86% of the genome...”

Response: Thank you for the suggestion. We have revised the sentence to start a new sentence for the “86% of the genome” part, as recommended.

Lines 106-108: Manual annotation to improve BUSCO scores is somewhat misleading, and it is unclear exactly how this was done. Suggest including a sentence to say something like: ‘This indicated that the genome coverage is high, but that some gene annotations are lacking’. Additional detail needs to be added to methods.

Response: Thank you for the insightful comment. We used BLASTp to manually identify the missing BUSCOs, following the approach outlined by Moggioli et al. (2023). This is mostly to confirm that our genome is relatively complete, and the higher BUSCO score after manual BLAST achieved that goal.

To annotate the *T. maxima* genome, we utilized comprehensive transcriptome data from various tissue types. Additionally, running BUSCO in genome mode yielded similar scores to those obtained in proteome mode on the annotated genes, suggesting that the issue is unlikely to be due to annotation issues. Given the high coverage of the *T. maxima* genome, and the fact that BUSCO scores for other chromosome assembly *Tridacna* genomes are also around 80%, the relatively lower score is unlikely to be caused by low genome quality either. It could be related to unique characteristics of giant clam genomes, such as gene complexity or intron structure (Jauhal et al. 2021), which are likely influenced by the TE insertions.

Since we have demonstrated the high quality of our genome using other metrics, such as coverage and N50, we did not further investigate the definitive cause of the initial lower BUSCO scores, as it is not well aligned with our main theme. We suggest that investigating the relatively low BUSCO scores be left for future studies focusing on the unique gene structures of giant clams.

Moggioli, G. et al. Distinct genomic routes underlie transitions to specialised symbiotic lifestyles in deep-sea annelid worms. *Nat. Commun.* 14, (2023).

Jauhal, A. A., & Newcomb, R. D. (2021). Assessing genome assembly quality prior to downstream analysis: N50 versus BUSCO. *Molecular Ecology Resources*, 21(5), 1416-1421.

Revisions:

Line 118-119: “Manual blastp of the missing BUSCOs decreased this value from 9.4% to 4.7%, as 12 out of the 24 missing BUSCOs could be manually found (Supplementary Table 2).”

Line 438-439: “Gene completeness was assessed using BUSCO v.5.4.2⁸⁸ and manually curated following Moggioli et al (2023) using blastp⁸⁹.”

Line 223 onward: “our BUSCO analysis suggested faster rates of molecular evolution in the coding sequences of *T. maxima*”. How have you come to this conclusion? I can’t see anything about this in the results?

Response: This sentence has been removed from the new version of the manuscript. See our response above for more information.

Line 233 onward: “our analyses pinpointed to the genomic basis of giant clams' large shell size, and unique regulation of the calcification processes”. The basis for this statement is tandem duplication of genes and gene family expansions, and is problematic because 1) the tandem duplications are not dated, and therefore could pre-date evolution of the large shell, and 2) all of these genes and GO terms are multifunctional (none are shell-specific), so their expansion could quite easily be due to other functions.

Response: The tandem duplications analysis was removed from the new version of the manuscript. Instead, we used the duplications results to classify the genes found in expanded families in the CAFE analysis. See our response to the second comment.

Lines 399-400: More details required here; how was DNA extraction performed (this is notoriously tricky for molluscan tissue?). Was any size selection/inhibitor removal performed?

Response: Thank you for your comment. We have added more details to clarify the DNA extraction process in the method section:

Revisions:

Line 396-401: “DNA was extracted using the Qiagen mini kit (Qiagen, Hilden, Germany). Due to the presence of small DNA fragments in the samples, an unsheared DNA size selection was performed with a cutoff at 20 kb. The PacBio SMRTbell library for PacBio Sequel was constructed using SMRTbell Express Template Prep Kit 2.0 (PacBio, Menlo Park, CA, USA) using the manufacturer-recommended protocol.”

Figure 1 is not referred to in the text. ‘Tridacna maxima’ is misspelt in the figure.

Response: This is fixed.

Figure 2 – the text on the outside of the circle is unreadable at this size. Suggest reducing the size (width) of panel B and increasing A. Also, it’s quite hard to relate the x-axis to the dates given in the text (e.g., 3mya) given the log scale. Suggest making this easier – e.g., indicating important timepoints specifically.

Response: We thank the reviewer for this comment to improve Figure 2 clarity. We have followed their suggestions and have increased the size of panel A, marked the important date of 3Mya, and highlighted the marks of 10k, 100k, 1M and 10M years ago in the x axis of panel B. We also checked that the outside of the circle of panel A is readable in the high-quality version of this figure. Please see the revised Fig. 2.

Reviewer #2 (Remarks to the Author):

Review of Manuscript COMMSBIO-24-1281-T: "Photosymbiosis Shaped Animal Genome Architecture and Gene Evolution as Revealed in Giant Clams":

This manuscript by Li et al presents a new giant clam genome (*Tridacna maxima*), which the authors use to investigate the signatures that symbiosis creates in a genome. The genome itself appears to be high quality and will be incredibly useful for investigating photosymbiosis in giant clams and the transposable element connection to symbiosis is extremely interesting.

However, my primary difficulty with this manuscript is the interpretation of the results. I appreciate that the focus is on symbiosis, but this is one giant clam genome compared to just a few other non-symbiotic taxa.

I find the stories included in the discussion to be unsubstantiated in many cases (as highlighted in my comments below). I understand the focus on broad gene expansion and contraction, but there are a lot of assumptions based on blast annotation in the interpretation of the results, without any functional data to back them up.

Response: We thank the reviewer for the constructive reviews which helped us greatly improve the manuscript. We understand the concerns of using one genome to infer genomic consequences of photosymbiosis. Although the main focus of the current manuscript is *Tridacna maxima*, we have now included two additional *Tridacna* chromosomal level genomes in our comparative analyses. We have shown that the genomic features we hypothesized to be associated with photosymbiosis ecology persist in the other two genomes as well.

As for using "a few non-symbiotic taxa", we are constrained by what genomic resources are available. Currently there are just not that many full genomes available from non-symbiotic cardiid (same family as *Tridacna*) species.

We do understand that we discussed potential gene functions and adaptations based on genomic data only. We agree it is not ideal to infer ecological consequences without functional data, but it is currently challenging to conduct experimental manipulations on long-lived, non-model organisms like giant clams. It is also a common approach to infer genomic adaptations using genomic and transcriptomic data only. See the following papers as examples.

Sun, Jin, et al. "Adaptation to deep-sea chemosynthetic environments as revealed by mussel genomes." *Nature ecology & evolution* 1.5 (2017): 0121.

Lan, Yi, et al. "Hologenome analysis reveals dual symbiosis in the deep-sea hydrothermal vent snail *Gigantopelta aegis*." *Nature communications* 12.1 (2021): 1165.

Ip, Jack Chi-Ho, et al. "Host–endosymbiont genome integration in a deep-sea chemosymbiotic clam." *Molecular Biology and Evolution* 38.2 (2021): 502-518.

We have toned down some of the statements and removed others (tandem repeats, hox genes, etc). Some of the aspects though (such as immune innovation and TE insertion), have more support from research in other photosymbiotic systems (such as coral and anemones) and experimental studies. So, we are more confident about our interpretations. Please see our detailed responses below.

Second, the demographic history section in the results came out of nowhere for me. How do demographic history changes tell us more about how symbiosis impacts the genome? The discussion goes into this a bit more and I can kind of see the logic, but the connection to the genome signatures story is not very strong.

Response: We added text to introduce the Result paragraph on demographic history. In addition, we added text to the Introduction to better frame why we think looking into historical demography is relevant to show that photosymbiosis (as a result of genomic adaptations), allowed giant clams to colonize new environments while also intimately tying their demographic history to that of the expansion and decline of shallow marine reef habitats. We further develop this hypothesis in the discussion which was shortened and made more to the point.

Revisions

Line 61-70: “Giant clams originated and diversified in the Indo-west Pacific in warm shallow tropical seas and have always been restricted to these environments^{14,15}. Genetic evidence suggests that all giant clams are photosymbiotic, and this relationship is thought to evolved once in their common ancestor at ~ 27 mya (SD = 4.4)¹⁶, coinciding with the global expansion of modern coral reefs¹⁷. During this period, the emergence of shallow marine habitats dominated by other photosymbiotic organisms likely facilitated the evolution of photosymbiotic traits in giant clams by providing suitable environment and symbiont reservoirs. The genomic adaptations enabling Tridacninae to host photosymbionts likely lead to this subfamily’s radiation (6 mya¹⁶), allowing species and their populations to expand throughout the Indo-Pacific. These same innovations have also closely tied their demographic history and geographic distribution to the fate of coral reef ecosystems.”

Line 203-208: “Genomes keep records of evolutionary and ecological forces that shaped populations and can be used to reconstruct the demographic history of a species³¹. This work provides the first genome-wide demographic study of a giant clam using PSMC, a coalescent-based approach. We showed that the acquisition of photosymbiotic traits in *T. maxima* resulted in a demographic history closely linked to major geoclimatic events that also affected the growth and decline of shallow marine reef habitats.”

Finally, some of the methodological procedures need to be explained more sufficiently (more below).

Response: We have revised the text to clarify the methodological procedures. Please see details below.

Revisions:

Line 396-401: “DNA was extracted using the Qiagen mini kit (Qiagen, Hilden, Germany). Due to the presence of small DNA fragments in the samples, an unshered DNA size selection was performed with a cutoff at 20 kb. The PacBio SMRTbell library for PacBio Sequel was constructed using SMRTbell Express Template Prep Kit 2.0 (PacBio, Menlo Park, CA, USA) using the manufacturer-recommended protocol.”

Line 438-439: “Gene completeness was assessed using BUSCO v.5.4.2⁸⁸ and manually curated following Moggioli et al (2023) using blastp⁸⁹.”

Methods: detailed parameters or declarations of using default settings, as well as software versions have been added. We also shared our main scripts and commands on Github for clarity and reproducibility.

Introduction

Page 2, Lines 51-53: Why is this intriguing? I’m not following the connection here. And the use of the word limited is confusing. Does this mean there is some evidence that shows the opposite?

Response: Thank you for your comment. We find the higher proportion of transposable elements intriguing because traditionally, studies have focused primarily on protein-coding regions, often overlooking the potential significance of repetitive elements. It is also intriguing, because not all marine bivalves, or marine cardiids, or other marine invertebrates are known to have such higher proportion of TEs; this pattern is found more in symbiotic animals, which indicates some ecological implications. We used the word “limited” because this aspect has not been fully investigated. We revised the sentence to highlight that this underexplored aspect of genome architecture may play a significant role in maintaining symbiotic relationships or be influenced by symbiosis. We also changed the word “limited” to “emerging” to avoid confusion:

Revisions:

Line 52-55: “Intriguingly, emerging evidence show that photosymbiotic animal genomes appear to possess higher proportions of transposable elements compared to non-symbiotic relatives^{8,11}, suggesting that this underexplored aspect of genome architecture may play a significant role in maintaining symbiotic relationships or be influenced by symbiosis.”

Page 2, Lines 55-57: What evidence exists that their large size is due to photosymbiosis? There are certainly large animals without symbiosis and small animals with symbiosis. A clearer explanation would be helpful here.

Response: There are indeed large animals without autotrophic feeding, but we are not suggesting that photosymbiosis is the only mechanism for organisms to achieve large sizes here. There are also photosymbiotic organisms that are small (and single-celled), but animal size is a trait that is influenced by complex factors, including the organism’s evolutionary history, ecological niche, interactions with other organisms, etc. We did not intent to say that photosymbiosis will ultimately

lead to large sizes in other lineages. There are, however, decades of experimental studies which demonstrate that photosymbiosis does play at least a partial role in large sizes among *Tridacna*.

Giant clams are mixotrophic, exhibiting both high efficiency in primary or autotrophic feeding mode via entrained photosymbiont nutrient translocation and secondary or heterotrophic feeding via filter feeding and uptake of inorganic carbon, compared to non-photosymbiotic clams (Neo et al. 2015). The proportion of carbon deposited in tissues relative to that respired was higher in *Tridacna gigas*, the largest giant clam species, compared to non-photosymbiotic heterotrophic bivalves (Klumpp et al. 1992). While autotrophy was considered the major source of carbon to giant clams, the potential importance of heterotrophy to total energy needs was significant and changed with the size of clam. Heterotrophy via filter feeding contributed 65% of total carbon (to respiration and growth) in small clams, whereas large clams were not reported to rely on heterotrophy as much, with filter-feeding providing only 34% of carbon, indicating autotrophy energy supply is more important in large clams. The ability to harness two modes of feeding across life stages and potentially achieve higher growth rates over decades given this **dual flexibility** is thought to explain the large size of some giant clam species compared to other non-photosymbiotic bivalves.

Killam, D., Das, S., Martindale, R. C., Gray, K. E., Paytan, A., & Junium, C. K. (2023). Photosymbiosis and nutrient utilization in giant clams revealed by nitrogen isotope sclerochronology. *Geochimica et Cosmochimica Acta*, 359, 165-175.

Klumpp, D. W., Bayne, B. L., & Hawkins, A. J. S. (1992). Nutrition of the giant clam *Tridacna gigas* (L.) I. Contribution of filter feeding and photosynthates to respiration and growth. *Journal of Experimental Marine Biology and Ecology*, 155(1), 105-122.

Neo, M. L., Eckman, W., Vicentuan, K., Teo, S. L. M., & Todd, P. A. (2015). The ecological significance of giant clams in coral reef ecosystems. *Biological Conservation*, 181, 111-123.

Results

Page 3. Lines 106-108: This manual curation of only BUSCOs appears to be an artificial inflation of the BUSCO score given that non-BUSCOs would not change during this process. Perhaps clearer explanation of the goal of this would be helpful, as well as a bit of text explaining why those were not annotated correctly in the first place.

Response: Thank you for the insightful comment. We used BLASTp to manually identify the missing BUSCOs, following the approach outlined by Moggioli et al. (2023). This is mostly to confirm that our genome is relatively complete, and the higher BUSCO score after manual BLAST achieved that goal.

To annotate the *T. maxima* genome, we utilized comprehensive transcriptome data from various tissue types. Additionally, running BUSCO in genome mode yielded similar scores to those obtained in proteome mode on the annotated genes, suggesting that the issue is unlikely to be due to annotation issues. Given the high coverage of the *T. maxima* genome, and the fact that BUSCO

scores for other chromosome assembly *Tridacna* genomes are also around 80%, the relatively lower score is unlikely to be caused by low genome quality either. It could be related to unique characteristics of giant clam genomes, such as gene complexity or intron structure (Jauhal et al. 2021), which are likely influenced by the TE insertions.

Since we have demonstrated the high quality of our genome using other metrics, such as coverage and N50, we did not investigate the definitive cause of the initial lower BUSCO scores, as it is not well aligned with our main theme. We suggest that investigating the relatively low BUSCO scores be left for future studies focusing on the unique gene structures of giant clams.

Moggioli, G. et al. Distinct genomic routes underlie transitions to specialised symbiotic lifestyles in deep-sea annelid worms. *Nat. Commun.* 14, (2023).

Jauhal, A. A., & Newcomb, R. D. (2021). Assessing genome assembly quality prior to downstream analysis: N50 versus BUSCO. *Molecular Ecology Resources*, 21(5), 1416-1421.

Revisions:

Line 118-119: “Manual blastp of the missing BUSCOs decreased this value from 9.4% to 4.7%, as 12 out of the 24 missing BUSCOs could be manually found (Supplementary Table 2).”

Line 438-439: “Gene completeness was assessed using BUSCO v.5.4.2⁸⁸ and manually curated following Moggioli et al (2023) using blastp⁸⁹.”

Discussion

Page 5, Line 223: Faster rates compared to what?

Response: This sentence has been removed from the manuscript. Please see more information in the previous response.

Page 5, Lines 224-225: And yet, the BUSCO analysis found >90% of these genes without manual curation, which is pretty consistent to other mollusk genomes. If the rates of evolution are that fast, why were you even able to manually curate them?

Response: BUSCO score without manual curation is lower than 90%, it's 81%. Please see more information in the previous response.

Page 5, Lines 226-227: Is it possible that Cdx was just not picked up in the analysis? If you had to manually annotate BUSCO genes, it seems like it would be possible that Cdx might not be annotated through automated pipelines. If this was checked manually, that information should be added.

Response: Indeed, the Cdx gene was not annotated in the *T. maxima* genome. Following Reviewer #1 comment, we were able to annotate it using the Cdx sequence from *T. crocea*. Therefore, relevant text was deleted from the manuscript, and annotation of Cdx was added.

Page 6, Lines 230-231: How would a loss of developmental regulation lead to specialization? Please explain.

Response: This section has been removed from the manuscript, as we were able to annotate the Cdx gene in our reference genome.

Page 6, Lines 235-236: But chitin binding and calcium ion binding are not exclusive to shell matrix proteins. What else could these be relevant to in the giant clam?

Response: The tandem repetition analysis, including the GO enrichment, has been removed from the manuscript. In the new version, the duplication results were used to classify the genes found by the CAFE analysis in the four duplication categories: tandem, dispersed, segmental and proximal. We found that most genes found in the *Tridacna* specific expanded families were dispersedly duplicated (1,032 genes, 91.1%). Interestingly, literature on gene duplication indicated that “duplicates generated by transposable elements (TEs) can act as pseudo-TEs and amplify to multiple copies via transposition. The distinct copies are often dispersed on different chromosomes, which stands in contrast with the most frequent form of gene duplication induced by errors during DNA replication.” (Ma et al., 2023). This results further support our hypothesis that the expanded TEs in *Tridacna* genomes may play a role in driving genomic adaptations to *Tridacna* photosymbiosis. We have added new discussion text to incorporate this result.

Ma, H., Wang, M., Zhang, Y.E. and Tan, S., 2023. The power of “controllers”: Transposon-mediated duplicated genes evolve towards neofunctionalization. *Journal of Genetics and Genomics*, 50(7), pp.462-472.

Revisions:

Line 164-170: “Gene duplication analyses were conducted to characterize the nature of duplication for genes found in the expanded families. For *T. maxima*, most genes were dispersedly duplicated (3371 genes, 88.3%), followed by 248 (6.5%) genes in proximal duplications, 137 (3.6%) in tandem duplications, and 61 (1.6%) in segmental duplications. Similar patterns were found in expanded gene families shared by all three *Tridacna* species: 1,032 genes (91.1%) were dispersedly duplicated, 38 genes (3.4%) in proximal duplications, 28 (2.5%) in segmental duplications, and 25 (2.2%) tandemly duplicated.”

Line 350-353: More than 90% of expanded genes in *Tridacna* were dispersedly duplicated, which is a signature of TE-facilitated gene duplication⁷⁵, in contrast with the most common form of gene duplication induced by DNA replication errors.

Page 6, Lines 239-242: How exactly are the authors imagining that expansions and tandem repeats of genes may be responsible for shell size? Is this something that has been seen in other organisms? The expression patters of these genes have not been identified; how do they even know they are used for biomineralization at all in this species? It also isn't clear how these functions relate to symbiosis, which is meant to be the primary focus of the manuscript.

Response: Expanded families and duplicated genes are widely used in genome studies to explain the molecular mechanisms that characterize particular taxa. In theory, genes that are duplicated and gene families that are expanded in a species, may diversify their functions and are related to evolutionary adaptations in that species. Even though the expression patterns are unknown, the gene families that are expanded in *Tridacna* are indicative of the diversification of functions and adaptations in that specific gene families. Please see the following citations on functional rules of gene family expansion/contractions

Baroncelli, R., Amby, D. B., Zapparata, A., Sarrocco, S., Vannacci, G., Le Floch, G., ... & Thon, M. R. (2016). Gene family expansions and contractions are associated with host range in plant pathogens of the genus *Colletotrichum*. *BMC genomics*, 17, 1-17.

Zhang, W., Zhang, X., Li, K., Wang, C., Cai, L., Zhuang, W., ... & Liu, X. (2018). Introgression and gene family contraction drive the evolution of lifestyle and host shifts of hypocrealean fungi. *Mycology*, 9(3), 176-188.

Sackton, T. B., Lazzaro, B. P., & Clark, A. G. (2017). Rapid expansion of immune-related gene families in the house fly, *Musca domestica*. *Molecular Biology and Evolution*, 34(4), 857-872.

Page 6, Lines 244-Page 7, Line 279: These whole paragraphs feel very hand-wavy to me. Is there any evidence of a connection between these genes and symbiosis? The example given is in a mouse, which does not have symbionts, and the rest of the argument seems to me to just say that symbionts provide energy (which we have evidence for) and that means that they have enough energy to get big so they don't need as many CTRP genes. But again, the authors don't compare with any other symbiotic organisms, so I have no idea whether this is just specific to this giant clam species. The same feels true of the immune-related genes in the next paragraph.

Response: We understand the reviewer's concerns about lack of empirical evidence. However, we would like to point out that understanding genetic mechanisms in long-lived, non-model organisms can start with comparative genomic works, and inferred functions. And the "discussion" section is a place to provide hypotheses on these inferred patterns. We did not intend this to be the final conclusion of why these gene families were contracted or expanded. We have revised the text to reflect that these proposed mechanisms are hypotheses and need further testing.

In the example of CTRP, the only reason we were able to formulate any hypothesis at all is because this is a medically important gene in diabetes studies, therefore its function is meticulously studied in model organisms (mice) and assessed (knockout experiment). There are hundreds of other genes identified in this study, whose function we did not discuss here, because their functions were not investigated in any systems, or cannot even be annotated, not to mention studied in other photosymbiotic organisms. CTRP is used to control blood glucose levels by signaling various pathways to lower blood glucose. In giant clams, symbiont derived energy is passed into the host hemolymph (analogous to blood) in the form of glucose. And since symbionts under light are constantly transporting glucose, it is plausible that a mechanism that is sensitive to high blood glucose level (and brings it down) has no fitness value and is therefore lost. We don't think this connection is too far stretched.

As for immune related genes, there are actually a body of evidence from cnidarians and other bivalves showing photosymbiotic host generally exhibit suppressed immune systems and down-regulate immune related genes, and we cite them in the text. We unfortunately can not perform functional assessment on those genes, but we think suggestions on their potential involvement in symbiosis are valuable in the discussion.

We have revised the CTRP discussion to clarify our logic:

Line 242-251: The striking contraction of the C1q/Tumor Necrosis Factor-Related Protein (CTRP) gene family in *T. maxima* (and in the other two *Tridacna* species) may also be linked to its symbiont-fueled growth and ultimately large size. CTRP has been well studied in mammalian models because it plays a significant role in body weight control through reducing blood glucose and insulin levels. For example, CTRP9 knockout mice models exhibit increased body weight and impaired insulin resistance⁴⁸. Giant clams derive substantial energy from their symbionts in the form of glucose⁴⁹. Given that *Tridacna* receives constant glucose supply from algal photosynthesis, it is possible that the glucose-sensitive CTRP gene family is no longer beneficial and is selected out to avoid interference with symbiont nutrient transfer. And the loss of CTRPs may in turn lead to less prohibited weight gain and growth.

Page 7, Lines 291-292: Long compared to what? And I would argue this is a hypothesis based on these data and prior information, not support of an assumption.

Response: We have removed the term “long” as it was unnecessary and caused ambiguity. This clarifies the comparison. We also agree that this is better presented as a hypothesis rather than support of an assumption. To address this, we have rephrased the sentence to reflect that the link between cilium-related genes and the photosymbiotic trait of *T. maxima* is a hypothesis based on additional evidence:

Revisions:

Line 299-303: “In addition, comparative transcriptomics studies found that cilium-related genes are upregulated in normal symbiotic conditions compared to when symbiosis is disrupted by darkness⁶⁰. This suggests that assumption that the cilium-related genes found exclusively in *T. maxima* may be linked to the unique photosymbiotic trait of this species.”

Page 7, Lines 296-297: What % and how similar?

Response: Thank you for your comment. We manually checked the functions of the expanded gene families and novel genes, and we observed similar functional terms appearing in both datasets (as shown in the Table 1 and supplementary table 1). Given the divergence in gene functions and the lack of uniform descriptions, quantifying this overlap as a percentage is challenging. However, we consistently observed representative functional groups in both, particularly those related to immune response, calcium regulation, and metabolism.

Page 7, Lines 306-308: This sentence comes out of left field for me. How does this relate to expanded gene families? There seem to be steps missing between nutrient benefits of photosymbiosis, positive selection, and gene family expansion. The connections are not clear to me based off of this text.

Response: Thank you for pointing this out. We agree that the connection between nutrient benefits, positive selection, and gene family expansion is not clear. We restructured our discussion and this part was deleted.

Page 8, Lines 335-341: I am confused by this paragraph. Suppression of immune systems and inhibition of host immune pathways are individual interactions. How are these interactions expected to lead to TE expansions across evolution?

Response: Thank you for your comment. 1) The immune system plays a key role in silencing TEs. In cases where immune pathways are suppressed or modulated, such as during symbiotic interactions, TEs may become more active and insert and replicate themselves into the genome. 2) Additionally, TEs are sometimes introduced into the genome by viruses. A weakened immune system or suppressed immune response could lead to increased viral infections or TE mobilization, resulting in more TEs being introduced and inserted into the genome. Over time, these inserted TEs can accumulate across generations, contributing to the overall expansion of TEs in the genome.

Methods

Page 9, Line 403 (and beyond): I'll use WTDBG2 as an example, but this goes for programs used across the pipeline; what parameters were used? If it was just defaults for everything, please provide that information. If not, the specific parameters used should be listed somewhere either in the main text or a supplement.

Response: Thank you so much for the suggestion. Detailed parameters or declarations of using default settings, as well as software versions have been added. We also shared our main scripts and commands on Github for clarity and reproducibility.

Page 10, Lines 418-419: Were these two transcriptomes merged with the generated assembly or used separately? Please clarify.

Response: The two transcriptomes were merged with our transcriptome and used to train Augustus. We have added these details.

Revisions:

Line 420-421: "Two additional published transcriptomes^{16,81} were merged with our new transcriptome to facilitate genome annotation."

Page 10, Lines 424-427: Why were these not included in the initial Augustus training? It would likely have been helpful to include as much RNAseq data as possible.

Response: We used a two-step annotation pipeline. In the first step, we trained Augustus with three transcriptomes. Augustus integrates the RNAseq information to help predict where introns and exons are found. There is no evidence that using more transcriptome data will improve Augustus predictions on gene features. Hence, we only used three transcriptomes for this first step. In the second step, we used all available *T. maxima* RNAseq reads to help improve these annotations from the first step, including a few new genes that were not picked up by Augustus and to improve the 5'/3' sequences. Details were added to the manuscript for clarification. See the details in the response below.

Page 10, Lines 430-434: Were these compared to gene models from MAKER? It isn't clear to me why these steps were only performed on a subset of gene models from the RNAseq data but not on all gene models.

Response: Gene models from RNAseq data were predicted following the standard pipeline: mapping with STAR, transforming to gff with StrigTie, and gene prediction with TransDecoder. Then, MAKER and TransDecoder annotations were merged using StringTie to obtain the final improved annotations. Details were added in the methods section for clarification.

Revisions:

Line 426-438: “Subsequently, all *T. maxima* RNAseq reads available at the NCBI SRA database (48 libraries, including different tissue samples: muscle, mantle, visceral mass, kidney, gonads, gills and byssus, see Supplementary Table 8) were used to improve the *ab initio* genome annotation. RNAseq reads were downloaded using sratoolkit v.3.0.0 (<https://github.com/ncbi/sra-tools/wiki/01.-Downloading-SRA-Toolkit>) and mapped to the genome assembly using STAR v.2.7.4a⁸⁵. Gene transfer files (GTFs) were generated from each individual bam file and subsequently merged using StringTie v.2.2.1⁸⁶. Gene models from the RNAseq data were predicted using TransDecoder v.5.6.0 (<https://github.com/TransDecoder/TransDecoder>), with homology searches to annotate and retain open reading frames (ORFs) with functional significance. BLASTP v.2.10.0⁸⁷ and hmmscan v.3.3 (<https://hmmer.org>) were used for homology searches against Uniref90 and Pfam databases, respectively. TransDecoder and *ab initio* MAKER annotations were merged using StringTie to obtain the final genome annotation.”

Page 10, Lines 434-435: How were duplicates in the annotation handled when merging gene prediction annotations?

Response: We used StringTie --merge to merge gene prediction annotations coming from MAKER and TransDecoder, giving the MAKER gtf file as the guide using the -G option. In this mode, StringTie merges the gene annotations into a non-redundant set of gene annotations, assembling the transfrags of the input GTF files from TransDecoder with the reference annotations from MAKER. Hence, this gave us all the annotations from MAKER plus some extra genes only

annotated with the RNAseq annotation pipeline. These details were added to the new version of the manuscript. See the revisions above.

Page 10, Lines 456-457: But which proteins? From both sets of gene models (MAKER and RNAseq)?

Response: Using the merged set of gene models. Once the annotations from MAKER and TransDecoder were merged, the merged set of gene predictions was used for all the rest of analyses. We have clarified this in the new version of the manuscript. See the response above.

Page 11, Line 460: Why the focus on tandem duplications? There are likely many other types of duplications present in the genome, but these would not be accounted for in this analysis unless I'm mistaken. Is there a specific reason for this choice?

Response: We have removed this section from the new manuscript. Now, we have used the gene duplication results to characterize the genes that were found by CAFE as expanded families in *Tridacna*.

Page 11, Line 480: An explanation of the metric used to determine significance is necessary here.

Response: We have added the metric used to determine significance: $p < 0.05$:

Revisions:

Line 469-470: "Significant ($p < 0.05$) expanded and extracted families were selected by a custom bash script."

Page 11, Lines 481-483: How exactly was this done? Were the same proteomes used? With what parameters? At what levels were the taxonomically restricted genes isolated? Just the species level? Also, it certainly would be possible that OrthoFinder would result in different gene families or clusters than OrthoDB, so it would be helpful to describe how gene families were confirmed across the two pipelines in terms of their membership.

Response: Thank you for your comment. We have clarified the methodology used. Specifically, the same proteomes were used with OrthoFinder, applying default parameters to assign genes to different gene families. Taxonomically restricted genes were defined as those gene families that contained only genes from *Tridacna*, or unassigned genes from *Tridacna*. We have added this clarification in the text.

You are absolutely correct that OrthoFinder and OrthoDB may produce slightly different gene families. However, we are using them for different purposes. OrthoDB contains pre-assigned annotated gene families but does not create new gene families specific to *Tridacna*. This is why we use OrthoDB to investigate gene family contractions, which allows us to leverage well-annotated gene family functions from OrthoDB. On the other hand, OrthoFinder allows us to

identify gene family expansions and taxonomically restricted genes specific to *Tridacna*. Since we are using these two methods to investigate different aspects of the genome, there is no need to cross-reference the results from the two pipelines. Please see the details of revised methods in line 456-480.

Figures

Figure 1: I don't find this figure very helpful. What is the main point I am supposed to take from it? The authors do not even investigate morphological adaptations or the immune system really, and one could argue that coding regions are divergent in every species.

Response: The figure is aimed to provide a summary of major genomic characters that are potentially driven by *Tridacna maxima* photosymbiosis ecology. We found it helpful for readers who are not familiar with genomic analyses and are more interested in the ecology of the species. We have modified it so there are no misunderstandings that this work includes morphological or experimental studies.

Figure 6: This figure suggests that many of the major expansions in *Tridacna* likely emerged prior to the origin of stable photosymbiosis. I also just realized that the authors use size for this rather than percentage of the genome. Given that the genome is just larger, would it not be useful to provide information on both? For example, in the table it is obvious that *Tridacna* has a much higher percentage of DNA Transposons and Rolling-Circle Transposons compared to *Cerastoderma*. It would be interesting to know at what point in time those differences emerged.

Response: Thank you for your comment. While we agree that percentage might be informative, determining the total genome size at different evolutionary time points is challenging, as the genomes expanded over time. Using a fixed percentage could lead to misleading trends, as it would not accurately reflect the changing genome sizes across different points in evolutionary history.

The current plot clearly illustrates the timing of key changes in transposon activity. For instance, the rate of RC transposon insertions remained consistently higher starting around 120 MYA. In contrast, DNA transposon activity was similar to that of *Cerastoderma edule* around 120 MYA but peaked around the time stable photosymbiosis emerged in *Tridacna*. These trends provide valuable insights into the timing of TE expansions, without relying on potentially misleading percentages. To further confirm the trends, we also included two more *Tridacna* species in the analyses. Please see the updated figure 5.

Reviewer #1 (Remarks to the Author):

The authors have made corrections that have significantly improved the manuscript (especially the synteny analysis and comparison to other *Tridacna* genomes, which are both very interesting). I believe some corrections are still required before the manuscript is acceptable for publication.

Response: Response: We really appreciate the comments, which are highly constructive. We have revised our manuscript as suggested. Please see the detailed response below.

I find that the claims in the abstract and conclusion are still too overstated. The authors pointed out in their rebuttal that ‘in order to definitively link photosymbiosis to any genomic features, one needs to either conduct genetic manipulations on the organism to see how they disrupt photosymbiosis, or use multiple symbiotic/non-symbiotic sister groups to test if certain genomic features are consistently associated with the symbiotic lineage. The former method is very challenging for long-lived non-model study systems like giant clams’. I completely understand this, however this does not remove the need for caveats and caution in interpretation. For example, in the abstract:

Line lines 24-25: ‘highlighting how abiotic and biotic factors dictate *T. maxima* microevolution’ needs to be rephrased to “revealing how abiotic and biotic factors may dictate *T. maxima* microevolution’

Line 25: “Comparative analyses revealed unique symbiosis-driven genomic features” needs to be rephrased to “Comparative analyses revealed genomic features that may be symbiosis-driven”.

Similar corrections need to be made to many sentences in the conclusion, too.

Response: Thank you very much for your valuable feedback. We have revised the two examples as suggested. Additionally, we have made further edits to the conclusion to ensure our statements are less definitive by using words such as “could” and “may.” Below are some examples of these revisions:

Revisions:

Line 466: We showed that symbiotic associations may impact animal genome evolution, at both gene and structural levels.

Line 467-468: We demonstrated that various aspects of the *T. maxima* genome such gene family and repetitive element evolution, could be influenced by their photosymbiosis ecology.

The analysis of ‘taxonomically restricted genes’ is problematic, as it appears to have been conducted through a comparison of *T. maxima* with other non-Tridacnid genomes. This is odd, as other parts of the manuscript include the two other *Tridacna* genomes. If the main question is about

potential links to photosymbiosis, why wouldn't you just compare *Tridacna*-specific genes to other non-photosymbiotic genomes, rather than the 2-step analysis conducted here? This creates issues in the discussion, for example, the statement 'A unique set of TRGs only found in *T. maxima* are related to ciliary structure and function' implies that these genes are not found in the other *Tridacna* genomes, which is not really what you mean. In my opinion it would be best if this analysis were redone to include all three *Tridacna* genomes, however I understand that this is a significant amount of additional work. Otherwise, careful explanation and rephrasing of the relevant sections is required to resolve the issue.

Response: Thank you for the comments. We indeed conducted TRG detection on all three *Tridacna* species as the reviewer suggested, and we highlighted genes uniquely found in all three *Tridacna* species but absent in other non-tridacnid bivalves in Figure 4B and all genes can be found in Supplementary Table 10. This approach was stated in the methods: "Analyses on the three *Tridacna* species were conducted similarly to those performed on *T. maxima*, though the annotation of expanded gene families differed.", and the results "Similarly enriched processes related to transportation, immunity, and metabolism are found in the genes and gene families uniquely shared by *T. maxima*, *T. crocea*, and *T. gigas* (Supplementary Table 1, Supplementary Table 10)." The reason we presented the three species TRG results only in supplementary and focused on *T. maxima* TRG for discussion is due to genome annotation (identifying genes from the genome) quality of the other two *Tridacna* species. Unlike other species with limited RNA data, the *T. maxima* genome annotation incorporates RNA libraries from seven different tissues—muscle, mantle, visceral mass, kidney, gonads, gills, and byssus—across 48 libraries. In contrast, the genomes of *T. gigas* and *T. crocea*, sequenced as part of the Aquatic Symbiosis Genomes project, were annotated using only one library in their standard pipeline, therefore do not have a comprehensive annotation of the genome, in other words, many genes in the genomes might not be identified because the lack of gene expression data. Including these two genomes made the TRG results more conservative, and risked ruling out potentially informative genes due to lack of annotation in the other two species.

In the discussion of *T. maxima* TRGs, we highlighted genes related to ciliary structure and function, which were also identified as functionally significant in another photosymbiotic bivalve (Li et al. 2024), and supported by histological evidence of *Tridacna gigas*. We therefore think these genes may play an important role in giant clam photosymbiosis and are worth discussing in this paper. The absence of these genes in the other two giant clam genomes may be linked to the quality of their genome annotations. To avoid confusion, we have rephrased these points for clarity.

Li, R., Zarate, D., Avila-Magaña, V., & Li, J. (2024). Comparative transcriptomics revealed parallel evolution and innovation of photosymbiosis molecular mechanisms in a marine bivalve. *Proceedings B*, 291(2023), 20232408.

Revisions:

Line 140-141: Thus, we conducted a comparative analysis of gene family evolution between *T. maxima* and 13 other non-*Tridacna* molluscan species.

Line 170-173: It is worth noting that these represented an underestimation of unique genes at the *Tridacna* genus level due to less extensive genome annotation of *T. crocea* and *T. gigas* – many

genes might not be identified from these two genomes because they were annotated by limited RNA libraries^{27, 28}.

Line 52: change to ‘emerging evidence shows’

Response: Revised

Line 62” change to ‘is thought to have’

Response: Revised

Line 81-82: ‘VHA’ is uninformative, change to ‘V-type H⁺ -ATPase’

Response: Revised

Lines 151-152: Unclear here whether the ‘other mollusc genomes’ includes the other *Tridacna* genomes, so are you discussing genes and gene families likely unique to *T. maxima*, or all *Tridacna*? Reading below it’s clear that you are not including the other *Tridacna* genomes, but this seems a strange way to go about the analysis (see comment above)

Response: Please see our third response above. We added “non-*Tridacna*” in front of the “other molluscan species” to clarify that one set of analyses only included *T. maxima*, and the other set includes three *Tridacna*.

Line 186: Change ‘known bivalve substitution rates’ to ‘calculated’ or ‘predicted’ bivalve substitution rates. Also, would you expect these to be the same in TE sequences, or elevated in comparison with the rest of the genome?

Response: Revised.

No TE-specific mutation rates have been estimated for Mollusca. Following the approach used in other Mollusca genome studies (e.g., Ip et al. 2021, Liu et al. 2021), we used gene substitution rates as the estimated TE sequence mutation rates. We expect that, once inserted, the substitution rates of TEs are similar to those in the rest of the genome. It is a common practice in genome papers. Please see some examples below.

Ip, J. C. H., Xu, T., Sun, J., Li, R., Chen, C., Lan, Y., ... & Qiu, J. W. (2021). Host–endosymbiont genome integration in a deep-sea chemosymbiotic clam. *Molecular Biology and Evolution*, 38(2), 502-518.

Liu, R. et al. De novo genome assembly of limpet *Bathycyba lactea* (gastropoda: Pectinodontidae): The first reference genome of a deep-sea gastropod endemic to cold seeps. *Genome Biol. Evol.* 12, 905–910 (2021).

Line 189: I’m assuming this should be a reference to Fig. 5.

Response: Revised

Line 205-208: The wording is too strong here – given that photosymbiosis was thought to have originated in stem Tridacninae at ~27mya, and that your demographic history inference only extends back to 3mya, you can't link the acquisition of photosymbiotic traits to anything, really.

Response: Thank you for the feedback. We did not intend to imply that the acquisition of photosymbiotic traits is linked to major geoclimatic events. Instead, we meant that being photosymbiotic influences the demographic history of giant clams, making it affected by major geoclimatic events. We have removed the phrase “acquisition of” to avoid confusion.

Line 208-216: Again, this is all too strongly worded. Your analysis is a model/estimate only, and you cannot definitively link historical climatic events to the (estimated) population size, you can only say that your estimates are congruent with these hypotheses.

Response: We have revised those sentences to include terms such as “probably,” “likely,” and “estimated,” to soften the tone and make it less definitive.

Revisions:

Line 221-229: The Pliocene extinction of marine and terrestrial megafauna, triggered by Northern Hemisphere Glaciation (NHG)^{32,33}, likely led to a population expansion in *T. maxima* (effective population size (N_e) peak at 130k individuals) and other shallow water species like corals^{34,35} and oysters³⁶ by freeing habitable niches. From 1.5 to 0.4 mya, both *T. maxima* and coral populations³⁷ was estimated to experience a steep decline, potentially due to the Mid-Pleistocene Transition (MPT) and increased climate variability³⁴. After stabilizing during the steady climate of the Mid-Brunhes Events (~0.4 Mya)³⁸, *T. maxima* populations was estimated to decline at the onset of the Last Glacial Period. This population pattern is once again mirrored in other shallow marine species, including photosymbiotic corals^{34,35} that serve as habitats for giant clams.

Line 221: change ‘must have’ to ‘likely’

Response: Revised

Line 224: change to ‘is estimated to have occurred around the same time’

Response: Revised

Lines 225-226: change to ‘likely also made their survival’

Response: Revised

Line 228: replace ‘observed’ with ‘estimated’

Response: Revised

Lines 236-240: this small paragraph on genes potentially related to shell evolution is out-of-the-blue, given shells are not mentioned at all prior to this (likely needs some description of shell evolution in the introduction). It also highlights the major difficulty with this analysis; photosymbiosis is not the only trait that has evolved in this lineage, and gene family expansion/contraction can be associated with any of these traits. It's fine to speculate, but the language needs to reflect the uncertainty throughout.

Response: We added a sentence in the introduction highlighting that *Tridacna* possesses the largest and heaviest shells among all bivalves. In the results section, we also discussed gene family expansions and GO enrichment related to shell formation and calcification. Given that large size is one of the iconic features of giant clams, we believe it is important to include a brief discussion on shell formation genes, even though photosymbiosis is the main focus of our study.

We agree that linking gene or gene family characteristics to specific traits is one of the major challenges when working with *Tridacna* as a non-model system.

However, we can compare *Tridacna* genomes with those of non-symbiotic cardiid genomes and hypothesize that certain unique features may be linked to distinctive traits, such as photosymbiosis or large size. We then seek supporting evidence from the literature on other systems, including corals, anemones, and other bivalves, to strengthen our hypothesis. For instance, Calmodulin-A-like and EGF-like domain-containing gene families have been shown to be associated with calcification and shell formation in other bivalves (Wang et al. 2022, Shimizu et al. 2022). Therefore, we hypothesize that the expansion of these gene families might contribute to the large size of *Tridacna*. We have rephrased certain sentences to make our statements less definitive.

Wang, X., Li, C., Lv, Z., Zhang, Z. & Qiu, L. A calcification-related calmodulin-like protein in the oyster *Crassostrea gigas* mediates the enhanced calcium deposition induced by CO₂ exposure. *Sci. Total Environ.* 833, 155114 (2022).

Shimizu, K. et al. Evolution of Epidermal Growth Factor (EGF)-like and Zona Pellucida Domains Containing Shell Matrix Proteins in Mollusks. *Mol. Biol. Evol.* 39, 1–16 (2022).

Revisions:

Line 61-62: They possess the heaviest shells among all extant bivalves, with a recorded weight reaching up to 700 pounds.

Line 256-260: Our analyses revealed potential genomic basis related to giant clams' large and heavily calcified shells^{43,44}. We found significant expansions in gene families that regulate calcification and growth in bivalves, such as the Calmodulin-A-like and the EGF-like domain-containing gene families^{45,46}. Expansions of these biomineralization gene families might have driven diversification of gene functions and novelties responsible for giant clams' extraordinary growth rates and shell sizes⁴⁷.

Lines 242-243: I can't see anything related to this gene family in the results section – it's present in Fig 4, but if this is 'striking' it should be explicitly pointed out in the results text.

Response:

To highlight the contraction of the gene family, the following sentence is added in the results session:

Line 156-157: In particular, many gene families related to the C1q/Tumor Necrosis Factor-Related Proteins (CTRP) are contracted (Figure 4B).

Line 249: change to 'has been selected against'

Response: Revised

Lines 250-251: Delete the 'and' at the beginning of the sentence.

Response: Revised

Line 262: change 'highlights' to 'may reflect the'

Response: Revised

Lines 269-279: the text here is overly long, repetitive, and entirely speculative. Rewrite to make more concise.

Response: We have revised this part to make it more concise.

Revisions:

Line 296-302: The complex evolution of immune system gene families in *T. maxima* hints at a highly sophisticated adaptation strategy, where the immune system is in part specialized in recognizing and managing specific symbiotic partners (gene family expansions) while other aspects are being selectively suppressed (gene family contractions), likely facilitates the tolerance and long-term maintenance of symbionts by minimizing the host's immune reactions against them, and the selective downregulation of certain immune pathways could result in a compromised ability to recognize and respond to pathogens.

Lines 336-335: how is an increased susceptibility to viral infections relevant to the discussion of TE content?

Response:

Some common categories of transposable elements (TEs) are hypothesized to have originated from viruses. For example, LTR retrotransposons are thought to have evolved from retroviruses (Wells & Feschotte, 2020). Additionally, viruses can facilitate the horizontal transfer of TEs between species (Gilbert & Cordaux, 2017). We have revised the paragraph to make the logic clearer.

Wells, J. N., & Feschotte, C. (2020). A field guide to eukaryotic transposable elements. *Annual review of genetics*, 54(1), 539-561.

Gilbert, C., & Cordaux, R. (2017). Viruses as vectors of horizontal transfer of genetic material in eukaryotes. *Current opinion in virology*, 25, 16-22.

Revisions:

Line 368-374: Another hypothesized mechanism that explains TE expansions after photosymbiosis establishment is a suppressed immune system, which can be a result of the contracted immune gene families and immune pathway inhibitions by their Symbiodiniaceae symbiont⁷¹. Firstly, a suppressed immune system can be more susceptible to viral infections, which is a major source of TE insertion. Further, suppressed immune functions can lead to a reduction in TE silencing mechanisms, such as RNA interference⁶⁹ and DNA methylation⁷⁰. Altogether, the host genome may be more vulnerable to TE integration and lack common mechanisms to remove them once integrated.

Lines 336-337: how is symbiont suppression of host immune pathways relevant? Or is this an explanation of the immune system suppression mentioned in the first sentence of the paragraph? This paragraph needs restructuring to make the points more clearly.

Response: The immune system plays a crucial role in the suppression of transposable elements (TEs), as it helps to maintain genomic stability by recognizing and silencing these potentially disruptive sequences. Therefore, suppression of host immune pathways by Symbiodiniaceae symbionts may result in increased TE activity and insertions. We have revised the text to clarify this point further.

Revisions:

Line 368-374: Another hypothesized mechanism that explains TE expansions after photosymbiosis establishment is a suppressed immune system, which can be a result of the contracted immune gene families and immune pathway inhibitions by their Symbiodiniaceae symbiont⁷¹. Firstly, a suppressed immune system can be more susceptible to viral infections, which is a major source of TE insertion. Further, suppressed immune functions can lead to a reduction in TE silencing mechanisms, such as RNA interference⁶⁹ and DNA methylation⁷⁰. Altogether, the host genome may be more vulnerable to TE integration and lack common mechanisms to remove them once integrated.

Lines 348-349: Given the preceding sentence wouldn't you expect expansion of KRAB-ZFPs only? Argument needs further development.

Response:

Zinc finger proteins are transcription factors with characteristic finger domains that play a crucial role in DNA and RNA regulation (Li et al., 2022). However, a significant portion of this protein family remains functionally unannotated, especially in non-model organisms. KRAB-ZFPs exemplify how members of this family can acquire functions beyond their original role in TE silencing. Given structural similarities across the family, other zinc finger proteins may also exhibit such functional diversification. Due to the challenges of annotating genes from non-model species, functional characterization is often limited to conserved domains like the zinc finger, while specific subfamily details may remain unidentified. We observe a clear trend of ZFP expansion, which could logically be linked to their known roles. We use terms such as “may” to convey the uncertain nature of our interpretations.

Li, X., Han, M., Zhang, H., Liu, F., Pan, Y., Zhu, J., ... & Zhang, B. (2022). Structures and biological functions of zinc finger proteins and their roles in hepatocellular carcinoma. *Biomarker research*, 10, 1-13.

Revisions:

The *T. maxima* genome showed expansion of numerous zinc finger gene families, which may indeed be an effect of TE-induced genomic innovation.

Lines 355-356: Sentence needs revising to something like ‘Unlike (XYZ TEs), the DNA and RC families likely underwent...’

Response: Revised.

Lines 360-361: Why is this extremely challenging for non-model organisms? It doesn't require functional techniques, so should be no more difficult in non-model organisms than in model organisms

Response: To identify TE families comprehensively, it is necessary to use not only de novo TE libraries, such as those generated from RepeatModeler, but also well-curated, lineage-specific TE libraries. These curated libraries are derived from existing data and refined manually by experts (Goubert et al. 2022). Significant community efforts and meticulous manual curation have been invested to produce high-quality TE annotations of model systems (human, *Drosophila melanogaster*, maize etc.) (Ou et al. 2019), but it is not the case for non-model organisms. For instance, we utilized the Mollusca Repbase library, which of course is not as extensive as a maize library. We have adjusted the wording to remove stronger terms, replacing “extreme” with “more”.

Goubert, C., Craig, R. J., Bilat, A. F., Peona, V., Vogan, A. A., & Protasio, A. V. (2022). A beginner's guide to manual curation of transposable elements. *Mobile DNA*, 13(1), 7.

Ou, S., Su, W., Liao, Y., Chougule, K., Agda, J. R., Hellinga, A. J., ... & Hufford, M. B. (2019). Benchmarking transposable element annotation methods for creation of a streamlined, comprehensive pipeline. *Genome biology*, 20, 1-18.

Line 370: delete 'meticulous'

Response: Revised

Figure 1: 'expansion and contraction of metabolic genes', change to either 'gene families' or 'gene repertoire' (the genes themselves are not expanded or contracted)

Response: Revised.

Reviewer #2 (Remarks to the Author):

This manuscript by Li et al presents a new giant clam genome (*Tridacna maxima*), which the authors use to investigate the signatures that symbiosis creates in a genome. The genome is of high quality and will be incredibly useful for investigating photosymbiosis in giant clams, and I find the addition of the two other *Tridacna* genomes more compelling. Overall, I find the manuscript much improved. I have just a couple additional minor points below:

Response: Thank you for the constructive feedback, which has significantly helped us improve the manuscript. Please find our detailed responses below.

Page 6, Lines 242-251: Has this gene (or others like it) been found to be relevant to symbiosis in other organisms, like corals? It would be helpful to have some additional context here about other symbiosis systems.

Response:

To our knowledge, CTRP has been studied in mammalian models such as mice (Guan et al. 2020, Yang et al. 2021). The absence of CTRP9 in giant clams might indirectly influence weight gain and growth, although this is not specifically tied to their photosymbiotic nature. In other photosymbiotic organisms, larger body size is not always associated with symbiosis. For example, coral polyps, which also rely on photosymbionts, are typically only 1-10 mm in diameter. The loss of CTRP might be a unique feature in the giant clams.

Guan, H., Wang, Y., Li, X., Xiang, A., Guo, F., Fan, J., & Yu, Q. (2022). C1q/Tumor necrosis factor-related protein 9: basics and therapeutic potentials. *Frontiers in Physiology*, 13, 816218.

Yang, J., Zhao, D., Chen, Y., Ma, Y., Shi, X., Wang, X., ... & Yuan, H. (2021). Association of serum CTRP9 levels with cardiac autonomic neuropathy in patients with type 2 diabetes mellitus. *Journal of Diabetes Investigation*, 12(8), 1442-1451.

Page 7, Lines 301-303: Does that mean that the authors would expect the other *Tridacna* species to use different systems? It seems like a big deal that these genes were not found in the other *Tridacna* species. Or do the authors mean that these genes were not found in other mollusks? I'm a bit confused.

Response: Please see our third response to reviewer 1 above. We added “non-*Tridacna*” in front of the “other molluscan species” to clarify that one set of analyses only included *T. maxima*, and the other set includes three *Tridacna*.